# MSP: Probabilistically Consistent Multi-Scale Action Generation

**Zhixuan Lin** [1]   **Gengqi Liu** [1]   **Chao Zheng** [1]   **Gao Lin** [1]   **Jindong Yu** [1]   **Song Gao** [1]   **Fei Wang** [1]

## Abstract

In robotic imitation learning, accurately modeling the multimodality and temporal correlations of long-horizon action sequences remains challenging. Long-horizon tasks require preserving global task intent while executing precise low-level control; otherwise, local errors can accumulate and lead to failure. While recent coarse-to-fine autoregressive models have improved action generation, they struggle to maintain consistency across hierarchies, leading to suboptimal performance in long-horizon tasks. To address these shortcomings, we propose Probabilistically Consistent Multi-Scale Action Generation (MSP), a novel coarse-to-fine approach that promotes cross-scale consistency. MSP adopts a streamlined multi-scale design by directly downsampling in a continuous latent action space. A scale-wise autoregressive Transformer is used to generate semantic conditions at each scale, which guide a lightweight MeanFlow model to capture multi-scale latent distributions, enabling probabilistically consistent refinement across scales. Through extensive simulation and real-world experiments, including long-horizon, multi-task, and few-shot generalization settings, we show that MSP outperforms existing coarse-to-fine methods, achieving state-of-the-art performance with high efficiency. Project Page

## 1. Introduction

Learning from Demonstrations (LfD) enables robots to acquire skills directly from expert demonstrations (Argall et al., 2009; Urain et al., 2025), and serves as an effective paradigm for learning human manipulation behaviors. When humans perform tasks, they typically plan high-level action intentions first and then gradually refine them into

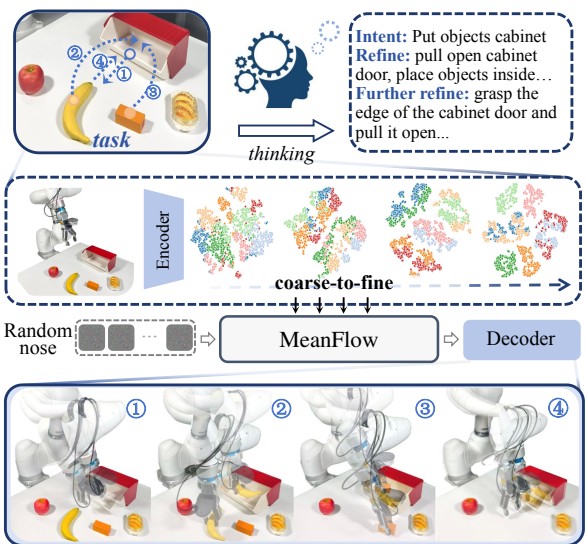

*Figure 1.* **Overview of MSP.** This multi-scale action generation approach enforces probabilistic consistency within a continuous latent action space. Inspired by human cognitive planning, MSP employs a hierarchical refinement strategy, enabling precise execution of long-horizon tasks.

precise movements (Ashe et al., 2006). This coarse-to-fine hierarchical decision-making process provides useful insights for robot policy learning. However, achieving such hierarchical generation in high-dimensional, continuous action spaces remains challenging: models must not only capture long-term dependencies and global structure, but also maintain sufficient inference efficiency and local precision for real-time control (Chi et al., 2023).

Existing work (Gong et al., 2025; Su et al., 2025) shows how coarse-to-fine generation mechanisms effectively capture temporal correlations in action sequences. Multi-scale representations based on discrete codebooks (Van Den Oord et al., 2017) improve long-term dependency modeling, but introduce quantization errors that can disrupt action continuity and fine-grained control. In contrast, autoregressive policies (Su et al., 2025; Zhang et al., 2025b) in continuous action spaces avoid discretization, but suffer from limitations in preserving coherent semantics across different hierarchies. Furthermore, diffusion models (Chi et al., 2023; Ze et al., 2024) and flow matching methods (Zhang & Gienger, 2024)

---

[1]Faculty of Robot Science and Engineering, Northeastern University, Shenyang 110000, China. Correspondence to: Fei Wang <wangfei@mail.neu.edu.cn>.

*Proceedings of the $43^{rd}$ International Conference on Machine Learning*, Seoul, South Korea. PMLR 306, 2026. Copyright 2026 by the author(s).

perform well in distribution modeling, but their generation relies on multi-step inference, making real-time control hardware-demanding. These limitations highlight that existing methods still struggle to reconcile action continuity, cross-scale consistency, and inference efficiency.

To address these challenges, we start from the inherent multi-scale nature of action sequences and present a core insight: an effective multi-scale action generation framework is capable of characterizing local dynamics at each scale, maintaining probabilistic consistency between different scales, and thereby enabling the natural and robust transmission of global intent to local control.

In this work, we propose **P**robabilistically Consistent **M**ulti-**S**cale Action Generation (**MSP**) for coarse-to-fine refinement in a continuous latent action space, as shown in Figure 1. MSP first encodes action sequences with a causal Transformer and constructs a multi-scale continuous latent action space via temporal downsampling. Then, given the observation embedding and latent representations from previous scales, a scale-wise autoregressive Transformer generates continuous semantics that condition the flow paths. Lastly, a lightweight scale-wise MeanFlow (Geng et al., 2025) model learns the average velocity field, enabling generation with a single function evaluation (1-NFE) that transforms noise into the target continuous latent representation for the current scale, thus modeling scale-wise probability distributions to maintain consistency during progressive refinement. Collectively, MSP effectively addresses the triplet of challenges of action continuity, cross-scale consistency, and efficient generation in long-horizon action modeling. Specifically, our contributions are as follows:

- MSP adopts a streamlined multi-scale design in a continuous latent action space for long-horizon behaviors, avoiding complex discrete tokenization and preserving fine-grained control.

- Conditioned on scale-wise autoregressive semantics, MSP models scale-wise latent action distributions, enabling probabilistically consistent coarse-to-fine refinement across scales.

- Through extensive evaluations on multi-task, long-horizon tasks, MSP achieves superior success rates, with up to a 6.8% improvement, despite using only 60% of the parameters of the state-of-the-art policy.

**Conflict of Interest Disclosure.** The authors declare no financial conflicts of interest.

## 2. Related Work

**Generative Imitation Learning.** Recently, generative modeling has achieved significant success in capturing complex action distributions. Unlike traditional behavior cloning (Pomerleau, 1988; Cui et al., 2022; Florence et al., 2022), generative methods focus on distribution modeling, which explicitly captures multimodal action distributions and sequential correlations within expert demonstrations. Discrete methods, such as VQ-BeT (Lee et al., 2024) and QueST (Mete et al., 2024), quantize action spaces into codebooks and use GPT-style (Brown et al., 2020) Transformers for sequence modeling. Alternatively, approaches such as Diffusion Policy (Chi et al., 2023) and its extensions (Ze et al., 2024; Lu et al., 2026) perform generative modeling directly in continuous action spaces through iterative denoising, but they often face real-time challenges. To enhance efficiency, ManiFlow (Yan et al., 2025) and FlowPolicy (Zhang et al., 2025a) introduce consistency constraints, while MP1 (Sheng et al., 2025) applies MeanFlow (Geng et al., 2025) to robot manipulation, achieving 1-NFE generation by learning interval average velocities. Despite these advances, these methods focus mainly on single-scale distributions, lacking explicit hierarchical modeling from global semantics to local control.

**Hierarchical Policies.** Accurately modeling the hierarchical semantics in actions is crucial for completing complex tasks (Triantafyllidis et al., 2023; Sun et al., 2025; Wang et al., 2025). ARP (Zhang et al., 2025b) uses action chunking within an autoregressive framework. Dense Policy (Su et al., 2025) adopts a bidirectionally expanded autoregressive policy to achieve precise predictions in the raw continuous action space. $H^3DP$ (Lu et al., 2026) uses multi-scale visual representations and hierarchical conditional diffusion. It tightly couples perceptual features with coarse-to-fine action generation. The work most similar to ours is CARP (Gong et al., 2025), which introduces multi-scale action representations within a discrete codebook (Zeghidour et al., 2021). Based on this representation, an autoregressive Transformer is employed for coarse-to-fine residual action prediction. Unlike CARP, our method does not rely on discrete tokenization or categorical cross-entropy loss. We perform multi-scale modeling directly in a continuous action space, leveraging coarse-to-fine scale-wise conditions derived from a scale-wise autoregressive model. We employ a scale-wise MeanFlow model to capture the probability distribution at each scale. Finally, a specialized MeanFlow loss is used to learn these scale-wise distributions, thereby avoiding the limitations introduced by discretization.

## 3. Preliminaries

We consider the problem of long-horizon robot behavior generation in continuous action spaces, where given a dataset of demonstration trajectories $\mathcal{D} = \{\tau_i\}_{i=1}^N$ with $\tau = \{(o_t, a_t)\}_{t=1}^H$, we aim to learn a policy $\pi(a_{t:t+T-1} \mid o_t)$ that produces executable action trajectories $a_{t:t+T-1}$

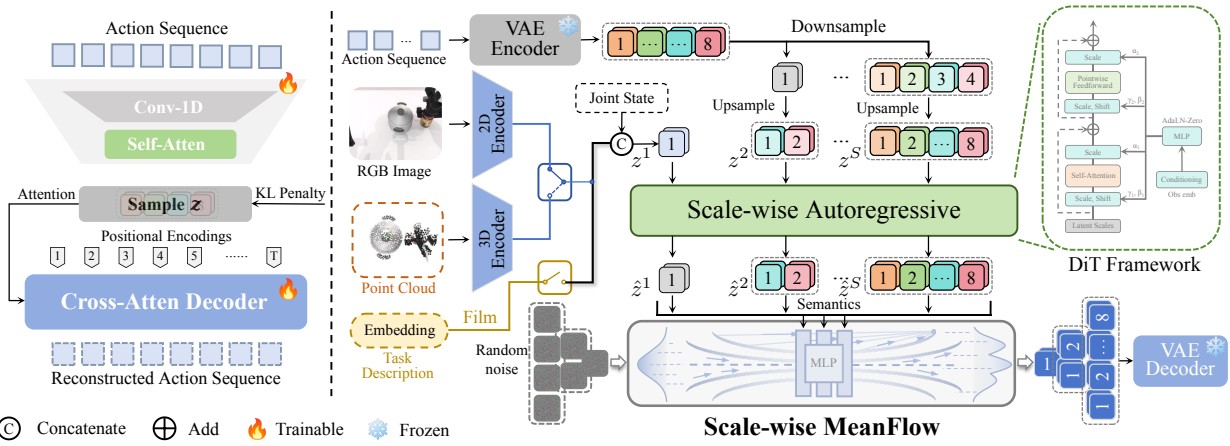

*Figure 2.* **Overview of the Two Stages of MSP. Stage 1:** MSP employs a causal Transformer within a VAE framework to encode long action sequences into a smooth, temporally structured continuous latent action space. **Stage 2:** Based on the encoded actions, MSP constructs a coarse-to-fine latent representation $\mathcal{Z} = \{z^1, \ldots, z^S\}$ via downsampling. Given the observation condition $C$, a scale-wise Transformer processes the sequence $\{C, \text{Up}(z^1, 2), \ldots, \text{Up}(z^{S-1}, 2)\}$, where $\text{Up}(\cdot, 2)$ denotes upsampling by a factor of 2, to generate semantic features $\hat{\mathcal{Z}} = \{\hat{z}^1, \hat{z}^2, \ldots, \hat{z}^S\}$ for each scale. These semantic features condition a MeanFlow model, which maps noise samples to the target latent at each scale using a single function evaluation (1-NFE). During inference, MSP recursively synthesizes the latent hierarchy in a coarse-to-fine manner and decodes the finest-scale latent into executable actions.

from the current observation $o_t$. Long-horizon tasks require preserving global task intent while executing precise low-level control. Motivated by the inherent multi-scale nature of action sequences, we adopt a coarse-to-fine modeling perspective and construct multi-scale temporal representations in a continuous latent action space via temporal downsampling. Let $C$ denotes the observation condition and $z^S$ denotes the finest-scale latent action trajectory. For a coarser scale $s < S$, we define $z^s = D_s(z^S)$, where $D_s(\cdot)$ is a deterministic temporal downsampling operator that maps the finest-scale latent trajectory to scale $s$. This deterministic relation induces the following coarse-scale distribution:

$$p_\downarrow(z^s \mid C) \equiv \int p(z^S \mid C)\delta(z^s - D_s(z^S)) \, dz^S. \quad (1)$$

Cross-scale probabilistic consistency requires the directly modeled scale-wise distribution $p(z^s \mid C)$ to be compatible with the downsampling-induced distribution $p_\downarrow(z^s \mid C)$, such that different temporal scales represent the same underlying action trajectory, with coarser scales capturing global intent and finer scales refining local temporal details. The detailed derivation is provided in Appendix A.

## 4. Method

This section details MSP, which promotes cross-scale consistency via probabilistically consistent multi-scale modeling in continuous latent action spaces, as shown in Figure 2. We first introduce continuous latent action modeling in Section 4.1, followed by probabilistically consistent multi-scale

modeling in Section 4.2.

### 4.1. Continuous Latent Action Modeling

Sequential actions are often highly correlated, making it crucial to model the skill semantics and temporal relationships in long action sequences. Unlike discrete quantization methods (such as FSQ (Mentzer et al., 2023) or VQ-VAE (Van Den Oord et al., 2017) ) which would disrupt the temporal continuity of actions, our method directly learns smooth and differentiable action representations by modeling Gaussian latent variable distributions (Kingma & Welling, 2013). Specifically, given a continuous action trajectory $a_{t:t+T-1} \in \mathbb{R}^{T \times D}$, we use linear projection and causal convolution to map it to the latent representation $A \in \mathbb{R}^{T' \times D'}$, where $T' = T/F$ is the downsampled sequence length with factor $F$. A causal Transformer is then applied to capture the temporal dependencies and local action structure of the sequence. Then, we use a linear layer to map it to the mean and variance $(\mu, \sigma)$ of a Gaussian distribution, thereby forming the variational posterior $q(z \mid a_{t:t+T-1}) = \mathcal{N}(z; \mu, \sigma^2 I)$. The latent variable $z \in \mathbb{R}^{T' \times d}$ sampled from this distribution captures the core motion intent of the corresponding action segment, and is decoded into the reconstructed action sequence $\hat{a}$. The training objective consists of a reconstruction loss and a KL regularization term:

$$\mathcal{L}_{\text{act}} = \|\hat{a} - a\|_1 + \beta \, D_{\text{KL}}\big(q(z \mid a) \, \big\| \, \mathcal{N}(0, I)\big), \quad (2)$$

where $\beta$ controls the KL regularization strength. We adopt a

small $\beta$, to balance latent action space smoothness and disentanglement with the diversity of action generation, thereby providing stable semantic representations for subsequent multi-scale policy optimization.

## 4.2. Probabilistically Consistent Multi-Scale Modeling

**Scale-wise Autoregressive Modeling.** Given a continuous action trajectory $\tau = \{a_1, \ldots, a_T\}$, autoregressive modeling represents the trajectory distribution by conditioning each action on all preceding actions, where the model is parameterized by $\theta$:

$$p(\tau) = \prod_{t=1}^{T} p(a_t \mid a_{<t}, \theta). \quad (3)$$

However, in long-horizon continuous control tasks, conventional autoregressive modeling often accumulates errors. To address this, inspired by FlowAR (Ren et al., 2025), we extend it to a multi-scale framework with coarse-to-fine latent sequence generation. Formally, we construct a set of coarse-to-fine multi-scale representations $\mathcal{Z} = \{z^1, z^2, \ldots, z^S\}$ by downsampling the highest-resolution latent action space $z$ in the time dimension, where $z^s = \text{Down}(z, r_s)$ with downsampling factor $r_s = 2^{S-s}$. Each scale $z^s$ has a dimension $t_s \times d$, where $t_s = T'/r_s$, and $d$ denotes the feature dimension of each latent token. On this basis, we decompose the joint distribution as:

$$p(z^{1:S}) = \prod_{s=1}^{S} p(z^s \mid z^{<s}, \theta), \quad (4)$$

where $z^{<s}$ denotes all latent representations at scales coarser than the current one. Each conditional distribution $p(z^s \mid z^{<s}, \theta)$ models the fine-grained variations at scale $s$ relative to coarser scales. Notably, our multi-scale token graph $\mathcal{Z}$ is directly downsampled from the highest-resolution latent action space $z$, without requiring complex multi-scale residual VQ-VAE designs. We set $S = 4$, with four scales and downsampling factors $r = \{8, 4, 2, 1\}$, making our scale design much simpler and more general.

**Scale-wise Autoregressive Semantic Generation for Flow Conditioning.** Given the latent variables of the previous coarser scales, $z = \{z^1, z^2, \ldots, z^{s-1}\}$, the Transformer generates the conditional semantics $\hat{z}^s$ for the current scale:

$$\hat{z}^s = T([C, \text{Up}(z^1, 2), \ldots, \text{Up}(z^{s-1}, 2)]), \quad (5)$$

where $T(\cdot)$ denotes the scale-wise autoregressive Transformer, $C$ is the observation condition as the initial scale input, and $\text{Up}(z, r)$ indicates that the latent variable $z$ is upsampled by a factor of $r$. We set $r = 2$, enabling coarser scale semantics to effectively guide the generation of finer scale temporal details. This generated output is denoted as

$\hat{z}^s$, acting as the semantic dependency for the $s$-th scale to condition the scale-wise MeanFlow module.

**MeanFlow Learning in Multi-Scale Latent Action Spaces.** Flow matching (Lipman et al., 2023) achieves generative modeling by learning a velocity field, which transforms the prior distribution into the target distribution. For each scale $s$, MSP extends flow matching to generate the scale latent $z^s$, and the interpolation state is expressed as $x_t^s = (1-t)z^s + t\epsilon$, where $\epsilon \sim \mathcal{N}(0, I)$. The instantaneous velocity field is $v_t^s = \epsilon - z^s$. In this work, we adopt MeanFlow (Geng et al., 2025), which models the average velocity field over the time interval $[r, t]$, defined as:

$$u(x_t^s, r, t) \triangleq \frac{1}{t-r} \int_r^t v(x_\tau^s, \tau) \, d\tau. \quad (6)$$

This design avoids the iterative integration process, enabling 1-NFE generation of the scale latent while maintaining distributional consistency. Since directly regressing the average velocity field from its integral definition is impractical, MeanFlow uses the "MeanFlow Identity" to transform it into an equivalent locally differentiable form:

$$u(x_t^s, r, t) = v(x_t^s, t) - (t-r)\frac{d}{dt}u(x_t^s, r, t). \quad (7)$$

The total derivative $\frac{d}{dt}u$ can be expanded as $v(z_t, t)\partial_x u + \partial_t u$. Based on this expansion, we convert this identity relation into a regressive supervised signal $u_{\text{tgt}}$:

$$u_{\text{tgt}} = v_t - (t-r)(v_t \cdot \partial_x u_\theta + \partial_t u_\theta). \quad (8)$$

This allows us to parameterize a network $u_\theta$ and encourage it to satisfy the "MeanFlow Identity". Specifically, we minimize the following objective:

$$\mathcal{L}_{\text{MF}}(\theta) = \mathbb{E}_{t,r,x,\epsilon} \|u_\theta(x_t^s, r, t) - \text{sg}(u_{\text{tgt}})\|_2^2, \quad (9)$$

where $\text{sg}(\cdot)$ denotes the stop-gradient operation. This removes the need for double backpropagation through Jacobian-vector products, avoids higher-order optimization, prevents gradient explosion, and improves training stability.

**Training Objective and Optimization.** At the $s$-th scale, conditioned on the scale-wise autoregressively generated semantic $\hat{z}^s$, the network $u_\theta(x_t^s, r, t \mid \hat{z}^s)$ is used to predict the average velocity field under. We incorporate Classifier-Free Guidance (CFG) within the MeanFlow framework. By weighting the conditional and unconditional velocity fields, this approach enables stronger semantic guidance. During training, the network is extended to a CFG-aware model $u_\theta^{\text{cfg}}(x_t^s, r, t \mid \hat{z}^s)$, with its loss defined as:

$$\mathcal{L}_{\text{cfg}}(\theta) = \mathbb{E} \left\| u_\theta^{\text{cfg}}(x_t^s, r, t \mid \hat{z}^s) - \text{sg}(u_{\text{tgt}}) \right\|_2^2. \quad (10)$$

The target average velocity $u_{\text{tgt}}$ is derived from the instantaneous velocity field under the following CFG scheme:

$$\tilde{v}_t^s \triangleq \omega v(x_t^s \mid \hat{z}^s) + (1-\omega)u_\theta^{\text{cfg}}(x_t^s, t, t), \quad (11)$$

where $v(x_t^s \mid \hat{z}^s)$ denotes the instantaneous velocity field conditioned on $\hat{z}^s$, and $u_\theta^{\text{cfg}}(x_t^s, t, t)$ represents the unconditional velocity. The guidance strength is controlled by $\omega$. In the multi-scale setting, CFG MeanFlow losses are computed independently at each scale and aggregated with scale-based weights. The overall training objective is:

$$\mathcal{L}_{\text{MSP}}(\theta) = \sum_{s \in \mathcal{Z}} \frac{s}{\max(\mathcal{Z})} \mathcal{L}_{\text{cfg}}^{(s)}(\theta) \,. \quad (12)$$

This weighting aligns the contribution of each scale with its latent size, balancing coarse semantic modeling and fine dynamic refinement, and improving the stability and consistency of multi-scale generation.

**Inference Pipeline.** Inference with MeanFlow replaces time integration with the average velocity field. Specifically, for any scale $s$ and state $x_t^s$, we compute the target latent $z^s$ via:

$$z^s = x_t^s - (t - r) u_\theta^{\text{cfg}}(x_t^s, r, t \mid \hat{z}^s) \,. \quad (13)$$

During inference, the scale-wise autoregressive Transformer first generates the initial semantic $\hat{z}^1$ based on the encoded observation conditions. This semantic $\hat{z}^1$ conditions the MeanFlow module, which converts noise samples into the target latent $z^1$ using Eq. (13), achieving 1-step sampling with $r = 0$ and $t = 1$. The generated latent $z^1$ is upsampled by a factor of 2 and combined with observations to feed the Transformer for the next scale. This process iterates until the final latent $z^S$ is obtained and decoded by the VAE decoder to generate the action sequence.

# 5. Experiments

In this section, we systematically evaluate MSP across simulated and real-world robotic tasks with varying temporal horizons. Through our experiments, we aim to answer the following questions:

1. How does MSP perform against state-of-the-art baselines on long-horizon and multi-task settings?

2. Does MSP learn reusable action semantics that enable effective multi-task and few-shot generalization?

3. How do the core design choices contribute to the performance and efficiency of our method?

4. Can MSP achieve robust real-world deployment?

## 5.1. Evaluation on Simulation Benchmarks

Across our simulation experiments, we evaluate MSP on two challenging benchmarks. We use LIBERO (Liu et al., 2023) to assess language-guided manipulation across multi-task imitation learning, few-shot transfer, and long-horizon scenarios. We further evaluate MSP on RoboTwin2.0 (Chen

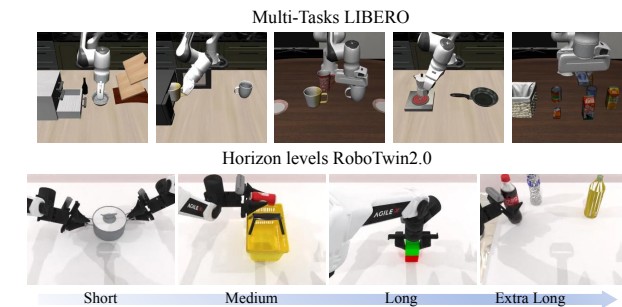

Multi-Tasks LIBERO

Horizon levels RoboTwin2.0

Short    Medium    Long    Extra Long

*Figure 3.* **Simulation Environments.** We evaluate multi-task imitation learning on the five LIBERO suites, and conduct evaluation on RoboTwin2.0 across four temporal horizon levels.

et al., 2025) to analyze performance under varying temporal horizons, selecting eight representative tasks grouped into four horizon levels. Figure 3 presents the simulated environments, while detailed task descriptions are deferred to Appendix B.

**Baselines.** We compare MSP with diverse baselines, including VAE-based ACT (Zhao et al., 2023), diffusion-based DP (Chi et al., 2023) and DP3 (Ze et al., 2024) with 3D representations, FlowPolicy (Zhang et al., 2025a), discrete autoregressive VQ-BeT (Lee et al., 2024) and QueST (Mete et al., 2024), latent space backward planning (LBP) (Liu et al., 2025), coarse-to-fine autoregressive CARP (Gong et al., 2025) and DSP (Su et al., 2025), and VLA models like OpenVLA (Kim et al., 2025), RDT (Liu et al., 2024), and $\pi_0$ (Black et al., 2026).

**Implementation Details.** We implement a unified action generation approach that seamlessly integrates 2D images, 3D point clouds, and language embeddings. In the LIBERO experiments, we adopt the same visual and language encoders as LBP, while for RoboTwin2.0 we follow the backbone designs used in DP and DP3. We use action chunks of length 32, which are compressed into latent sequences of length 8. The multi-scale token map $\mathcal{Z} = \{1, 2, 4, 8\}$ is then generated through scale-wise downsampling. We adopt DiT block (Peebles & Xie, 2023) for scale-wise autoregressive modeling, and implement MeanFlow with a lightweight three-layer MLP ResNet in place of a high-dimensional U-Net. Additional details are provided in Appendix C.

## 5.2. Performance of MSP on Long-Horizon Tasks

**RoboTwin2.0 Results.** As shown in Table 1, MSP exhibits increasingly pronounced advantages as task horizons grow. While performance is comparable across methods on short-horizon tasks, MSP consistently outperforms all baselines on long and extra-long horizon tasks, where many competing methods degrade significantly. Across eight tasks with varying horizons, MSP achieves an average success rate of 63.25% under 2D observations and 82.25% under 3D

*Table 1.* **RoboTwin2.0 Results.** Success rates on both image and point cloud-based inputs across varying horizons: **Short (100–150 steps)**, **Medium (170–260 steps)**, **Long (280–350 steps)**, and **Extra Long (450–650 steps)**. Models are trained on 50 clean demonstrations and averaged over 100 rollouts. **Tasks:** (1) lift pot; (2) move can pot; (3) place empty cup; (4) place can basket; (5) handover block; (6) stack blocks two; (7) stack bowls three; (8) put bottles dustbin. **Bold** indicates the best result, while underline denotes the second-best.

| Method | Obs. | Short | | Medium | | Long | | Extra Long | | Avg |
|---|---|---|---|---|---|---|---|---|---|---|
| | | (1) | (2) | (3) | (4) | (5) | (6) | (7) | (8) | |
| RDT | Img | 72 | 25 | 56 | 19 | 45 | 21 | 51 | 21 | 38.75 |
| $\pi_0$ | Img | 84 | 58 | 37 | 41 | 45 | 42 | 66 | 54 | 53.375 |
| ACT | Img | **88** | 22 | 61 | 1 | 42 | 25 | 48 | 27 | 39.25 |
| DP | Img | 39 | 39 | 37 | 18 | 10 | 7 | 63 | 22 | 29.375 |
| DSP | Img | 68 | 56 | 34 | 56 | **64** | 28 | 57 | 48 | 51.375 |
| **MSP** | Img | 85 | **61** | **68** | **64** | 56 | **44** | **72** | **56** | **63.25** |
| DP3 | PC | **97** | 70 | 65 | 67 | 70 | 24 | 57 | 60 | 63.75 |
| FlowPolicy | PC | 30 | 50 | 61 | 37 | 6 | 27 | 21 | 20 | 31.5 |
| DSP3D | PC | 83 | 61 | 57 | 54 | 80 | 34 | 84 | 60 | 64.125 |
| **MSP3D** | PC | 96 | **88** | **86** | **69** | **90** | **65** | **85** | **79** | **82.25** |

*Table 2.* **Performance on LIBERO Suites.** We report the average success rates over 50 rollouts per task. **Bold** indicates the best result, while underline denotes the second-best.

| Method | LIBERO | | | | |
|---|---|---|---|---|---|
| | Goal | Spatial | Object | Long | Avg |
| DP | 68.3 | 78.3 | 92.5 | 50.5 | 72.4 |
| FlowPolicy | 90.4 | 95.0 | 94.0 | 75.2 | 88.65 |
| QueST | 80.8 | 87.4 | 93.6 | 68.8 | 82.65 |
| CARP | – | – | – | 60.0 | 60.0 |
| LBP | – | – | – | 88.6 | 88.6 |
| OpenVLA | 79.2 | 84.7 | 88.4 | 53.7 | 76.5 |
| $\pi_0$ | **95.8** | 96.8 | 98.8 | 85.2 | 94.15 |
| DSP | 90.8 | 92.8 | 99.0 | 90.0 | 93.15 |
| **MSP** | 93.8 | **97.0** | **100** | **96.8** | **96.9** |

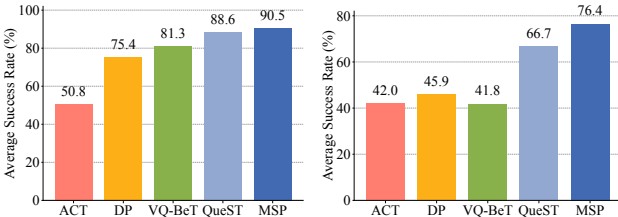

*(a)* Multitask on LIBERO-90. *(b)* Few-shot on LIBERO-Long.

*Figure 4.* **Multi-task and Few-shot Performance.** We evaluate the multi-task performance of MSP on LIBERO-90 and its few-shot generalization to unseen tasks.

visual inputs, yielding the best overall performance. These results demonstrate that MSP is robust to error accumulation and long-horizon dependencies, owing to coarse-to-fine multi-scale generation in continuous latent action spaces that enables global task structures to reliably guide local actions over long sequences.

**LIBERO Results.** Table 2 reports the average success rates across the four LIBERO suites. MSP achieves state-of-the-art performance with an average success rate of 96.9%, outperforming DSP and $\pi_0$ by 3.75% and 2.75%, respectively. The gains are most pronounced on LIBERO-Long, where MSP improves over DSP and $\pi_0$ by 6.8% and 11.6%, while slightly lagging behind on LIBERO-Goal. We attribute these improvements, especially on long-horizon tasks, to our multi-scale modeling in continuous latent action spaces, which preserves cross-scale probabilistic consistency via MeanFlow and enables robust coarse-to-fine propagation of global intent. In contrast, CARP relies on discrete codebooks, making it more susceptible to accumulated quantization errors in long-horizon settings.

**Generalization via Reusable Action Semantics.** We further evaluate MSP on multi-task imitation learning using LIBERO-90. As shown in Figure 4, our model achieves a 90.5% average success rate across 90 tasks, demonstrating strong performance. For few-shot generalization, we fine-tune MSP pre-trained on LIBERO-90 on unseen LIBERO-Long tasks using only 5 demonstrations per task, where MSP outperforms QueST by 9.7%. These results indicate that multi-scale continuous action modeling enables the learning of reusable high-level action semantics, leading to stronger generalization in low-data settings. While QueST uses discrete latent skills via finite scalar quantization, this design imposes hard semantic boundaries and limits flexibility for fine-grained motion adaptation to new tasks.

### 5.3. Impact of Core Design Choices on Performance

We conduct ablation studies on LIBERO-Long to analyze the contributions of key design choices in MSP.

**Ablation on Generative Components.** As shown in Table 3, mapping action sequences into a continuous latent action space is crucial for effective modeling in MSP. In long-horizon tasks, applying multi-scale generative models

*Table 3.* **Ablation on Key Components.** We analyze the impact of VAE latent design, scale configurations, and probabilistic modeling choices. The final setting is highlighted in gray.

| VAE | Scale | Diffusion | Flow-matching | Meanflow | Avg |
|---|---|---|---|---|---|
| | | ✓ | | | 79.6 |
| ✓ | ✓ | | ✓ | | 82.2 |
| | | ✓ | | | 79.2 |
| ✓ | | | | ✓ | 88.4 |
| ✓ | ✓ | | | ✓ | 89.6 |
| ✓ | ✓ | | | ✓ | 96.8 |

*Table 4.* **Ablation on Scale Configurations.** By varying the action length $T$ and downsampling factor $F$, we compare MSP across scale configurations. The final setting is highlighted in gray.

| Set | $T$ | $F$ | Scales | Avg |
|---|---|---|---|---|
| | 32 | 4 | [1, 2, 4, 8] | 96.8 |
| A | 32 | 4 | [1, 4, 8] | 91.6 |
| | 32 | 4 | [1, 8] | 91.6 |
| | 16 | 4 | [1, 2, 4] | 90.8 |
| B | 64 | 4 | [1, 2, 4, 8, 16] | 90.2 |
| | 64 | 4 | [1, 4, 16] | 79.4 |
| C | 32 | 2 | [1, 2, 4, 8, 16] | 89.6 |
| | 32 | 8 | [1, 2, 4] | 93.0 |

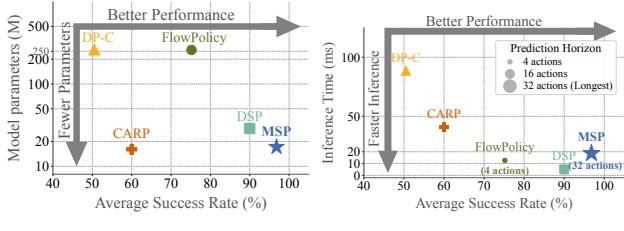

*(a) Avg. SR & Params*     *(b) Avg. SR & Inf. Time*

*Figure 5.* **Lightweight and Rapid Inference.** Comparison of Average Success Rate (Avg. SR), Action-Head Parameters (Params), and Inference Time (Inf. Time) on LIBERO-Long

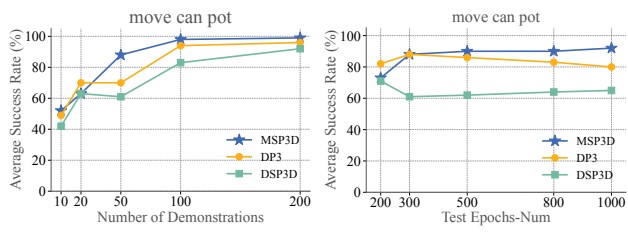

*(a) Demonstration Efficiency*     *(b) Learning Efficiency*

*Figure 6.* **Ease of Training.** On the *move can pot* task in RoboTwin2.0, we compare the learning efficiency and demonstration utilization efficiency of different models.

directly to the high-dimensional raw action space fails to reliably capture global structure. Introducing scale-wise autoregressive modeling leads to a clear performance improvement; notably, MeanFlow-based scale-wise flow matching significantly outperforms diffusion-based and standard flow-matching approaches, as it avoids the structural degradation caused by in multi-step generation. These results indicate that the performance gains of MSP stem from the synergy between a continuous latent action space, a scale-wise autoregressive structure, and 1-NFE MeanFlow generation, which effectively preserves multi-scale consistency.

**Ablation on Scale Configurations.** We systematically analyze the influence of action length $T$, downsampling factor $F$, and temporal scale design on long-horizon tasks. As shown in Table 4, continuous and dense multi-scale temporal modeling is crucial for capturing long-term dependencies: retaining all intermediate scales $[1, 2, 4, 8]$ with $T = 32$ and $F = 4$ achieves the optimal success rate of 96.8%. Removing intermediate scales or using sparse, non-continuous scales disrupts the coarse-to-fine semantic flow, causing more pronounced performance degradation for long sequences ($T = 64$). Ablation also shows that $F = 4$ balances noise reduction with the preservation of key semantic information, confirming that the advantages of MSP mainly come from continuous, stepwise multi-scale temporal modeling, which is critical for stable long-horizon execution.

### 5.4. Efficient MSP

We evaluate the action head of each method in terms of parameter count and single-step inference time, excluding the visual backbone. As shown in Figure 5, MSP uses only about 60% of the parameters of DSP while achieving better performance, and can generate 32 actions per forward pass compared to 16 for DSP, enabling longer execution per inference. This efficiency stems from the lightweight multi-scale latent action design and the 1-NFE MeanFlow generation. As shown in Figure 6, MSP improves steadily during training, whereas DP degrades and DSP recovers slowly, indicating stable training behavior. MSP also exhibits exceptional data utilization efficiency, rapidly improving performance with limited demonstrations.

### 5.5. Adapting MSP to Real-World Robots

**Real-World Experiments.** The real-world experimental setup is shown in Figure 7. We use a Doosan robotic arm with an RG6 gripper, teleoperated via a 3D Connexion space mouse, and equipped with three RGB cameras. An L515 camera is mounted on the wrist to provide fine-grained local observations, while two D435 cameras provide third-person views, including front and top-down views. For each task, we collected 50 human demonstrations. To evaluate the robustness and effectiveness of MSP in real-world scenarios, we designed five tasks with different temporal horizons: *Pick apple*, *Hang cup*, *Stack cups*, *Place object cabinet* and

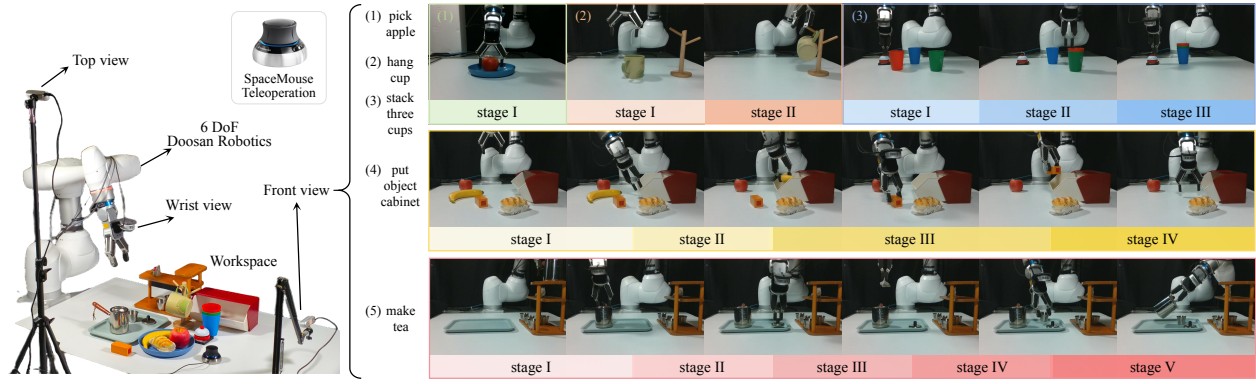

*Figure 7.* **Real-World Setup.** Left: The real-world experimental tabletop setup, consisting of a 6-DoF Doosan robotic arm equipped with an RG6 gripper, with three cameras providing multi-view observations; Right: Stage-wise trajectory visualizations for five tasks with different temporal horizons.

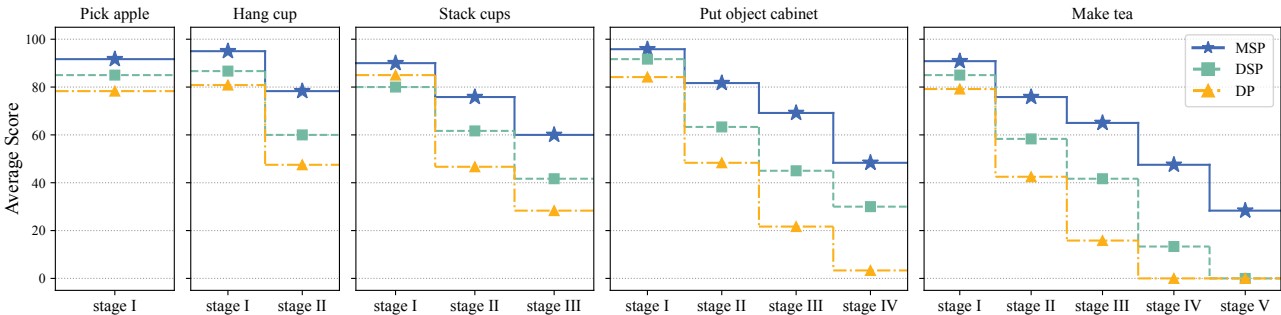

*Figure 8.* **Real-World Performance Across Task Horizons.** We compare DP, DSP, and MSP on five tasks with increasing temporal horizons. The results show that MSP consistently maintains higher execution stability and success rates as task stages and horizons grow, significantly outperforming baseline methods in long-horizon, multi-stage scenarios.

*Make tea*. Additional details are provided in Appendix D.

**Baselines.** We compare against DP and DSP. All methods use the same visual backbone as the DP.

**Metrics for Multi-Stage Tasks.** Following LBP (Liu et al., 2025), we adopt a stage-based scoring system to evaluate task performance, with 30 rollouts for each task. Each stage is assigned a discrete score from the set 0, 25, 50, 75, 100 to indicate its completion progress, with a full score of 100 awarded only when the stage goal is fully achieved. For multi-stage tasks, the next stage's score is only counted once the current stage is fully completed. This evaluation protocol effectively prevents partial successes from masking overall failure, providing a more accurate measure of the model's stability and completeness in multi-stage task execution.

**Real-World Experiment Results.** Figure 8 summarizes the real-world results, comparing methods across varying task temporal horizons. Notably, the performance gap between MSP and the baselines becomes increasingly pronounced as the number of task stages and execution horizons grow. In the single-stage *Pick apple* task, all methods achieve high success rates, demonstrating competence in short-horizon subtasks. In multi-stage tasks (*Hang cup* and *Stack cups*), the performance differences between methods begin to emerge. DP and DSP can typically complete initial stages like grasping and relocation, but they are more likely to fail in later stages such as precise alignment and successive stacking, leading to premature task termination. In contrast, MSP maintains robustness throughout these sequences, significantly outperforming the baselines.

This trend is further amplified in long-horizon, multi-stage tasks such as *Place object cabinet* and *Make tea*, which comprise complex sequences of subtasks. The success rates of DP and DSP drop significantly in later stages and may even fail completely, with failures mainly occurring during repeated grasping, placing, and realignment phases. MSP demonstrates higher execution stability, allowing tasks to progress consistently to completion. These results indicate that, by leveraging cross-scale consistent coarse-to-fine action modeling, MSP can stably generate longer continuous action sequences, thereby reducing discontinuities caused by frequent re-prediction, effectively mitigating error accumulation during long-horizon execution, and exhibiting stronger robustness in complex real-world tasks.

# 6. Conclusion

We present MSP, a novel and efficient coarse-to-fine approach that enforces cross-scale consistency. We show that probabilistically consistent coarse-to-fine generation in continuous spaces overcomes key limitations of long-horizon action modeling, including action continuity, cross-scale consistency, and efficiency. We further demonstrate that it enables learning reusable high-level action semantics, improving generalization in low-data settings. Extensive simulation and real-world experiments validate the superior performance of MSP on long-horizon tasks.

**Limitations.** MSP is motivated by the temporal correlation and shared skill semantics commonly observed in long-horizon action sequences. However, real-world sequences may also contain weakly related or unrelated segments, such as pauses, task switches, or behaviors from different sub-tasks, where a coherent coarse-to-fine temporal structure may be less suitable. Moreover, long-horizon real-robot deployment can still suffer from compounding errors, limited recovery capability after unexpected failures, and robustness degradation under distribution shifts in perception, object configurations, or dynamics.

**Future Work.** Future work could address these issues by incorporating task segmentation, uncertainty-aware execution, closed-loop replanning, explicit recovery mechanisms, and online adaptation in more diverse real-world manipulation scenarios. Beyond action generation, an important future direction is to extend this framework toward coarse-to-fine planning in visual latent spaces, enabling joint optimization of future video representations and action trajectories.

## Acknowledgements

This work was supported in part by the National Natural Science Foundation of China under Grant 62373087, in part by the Liaoning Provincial Applied Basic Research Program under Grant 2025JH2/101300009, and in part by the Liaoning Revitalization Talents Program under Grant XLYC24110114.

## Impact Statement

This paper presents work whose goal is to advance the field of Machine Learning. There are many potential societal consequences of our work, none which we feel must be specifically highlighted here.

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

## A. Derivation of Cross-Scale Probabilistic Consistency

This section provides the derivation of the cross-scale probabilistic consistency formulation in MSP. In our multi-scale latent action space, different scales correspond to different temporal resolutions of the same underlying latent action trajectory. Specifically, the finest-scale latent action sequence has the highest temporal resolution, while coarser scales are obtained by temporally downsampling this finest-scale representation. Therefore, latent variables at different scales are not independent random variables, but are coupled through deterministic temporal transformations.

Let $z^S$ denote the finest-scale latent action sequence, and define the scale-specific downsampling operator as

$$D_s(\cdot) = \text{Down}(\cdot, r_s). \tag{14}$$

Given the observation condition $C$, we assume that the finest-scale latent variable follows the conditional distribution $p(z^S \mid C)$. Since the latent variable at scale $s$ is obtained from the finest-scale latent variable through deterministic downsampling, we have

$$z^s = D_s(z^S). \tag{15}$$

Once $z^S$ is given, the value of $z^s$ is uniquely determined and does not introduce additional randomness. More specifically, for any measurable set $\mathcal{A}$ in the latent space of scale $s$, we have

$$\mathbb{P}(z^s \in \mathcal{A} \mid z^S, C) = \begin{cases} 1, & D_s(z^S) \in \mathcal{A}, \\ 0, & D_s(z^S) \notin \mathcal{A}. \end{cases} \tag{16}$$

This indicates that, conditioned on $z^S$, the conditional distribution of $z^s$ is a degenerate distribution whose entire probability mass is concentrated at $D_s(z^S)$. In the continuous-variable setting, a single point does not have an ordinary probability density. Therefore, the Dirac delta function is commonly used to represent such a distribution concentrated at a single point. For any test function $\phi(\cdot)$, the Dirac delta satisfies the sifting property:

$$\int \phi(z^s)\delta(z^s - a)\, dz^s = \phi(a). \tag{17}$$

Thus, $\delta(z^s - a)$ can be interpreted as a degenerate distribution that places all probability mass at $a$. By setting $a = D_s(z^S)$, the conditional distribution of $z^s$ given $z^S$ can be written as

$$p(z^s \mid z^S, C) = \delta\big(z^s - D_s(z^S)\big). \tag{18}$$

This term characterizes the deterministic relationship between two scales: the conditional distribution has probability mass only when $z^s = D_s(z^S)$. In other words, the Dirac delta acts as a constraint, or equivalently a filtering term, that preserves only those latent variables satisfying the downsampling relation.

According to the probability chain rule, conditioned on the observation $C$, the joint distribution of $z^s$ and $z^S$ can be written as

$$p(z^s, z^S \mid C) = p(z^S \mid C)p(z^s \mid z^S, C). \tag{19}$$

Substituting the deterministic conditional distribution above into this joint distribution gives

$$p(z^s, z^S \mid C) = p(z^S \mid C)\delta\big(z^s - D_s(z^S)\big). \tag{20}$$

Next, by marginalizing over the finest-scale latent variable $z^S$, we obtain the distribution at scale $s$ induced by the finest-scale distribution:

$$p_\downarrow(z^s \mid C) = \int p(z^s, z^S \mid C)\, dz^S. \tag{21}$$

Therefore,

$$p_\downarrow(z^s \mid C) = \int p(z^S \mid C)\delta\big(z^s - D_s(z^S)\big)\, dz^S. \tag{22}$$

Since $D_s(z^S) = \text{Down}(z^S, r_s)$, the above expression can also be written as

$$p_\downarrow(z^s \mid C) = \int p(z^S \mid C)\delta\big(z^s - \text{Down}(z^S, r_s)\big)\, dz^S. \tag{23}$$

The downsampling-induced distribution has a direct intuitive interpretation. Given the observation condition $C$, the finest-scale latent variable $z^S$ follows the distribution $p(z^S \mid C)$. When each possible $z^S$ is mapped to scale $s$ through the deterministic downsampling operator $\mathrm{Down}(\cdot, r_s)$, it yields a corresponding coarse-scale latent variable. Since multiple different finest-scale latent trajectories may be mapped to the same $z^s$ after downsampling, the probability of $z^s$ should be determined by all finest-scale trajectories that can be downsampled to $z^s$. In the integral above, the Dirac delta term selects all finest-scale latent trajectories satisfying $\mathrm{Down}(z^S, r_s) = z^s$, while the integral accumulates their probability mass under $p(z^S \mid C)$. Therefore, $p_\downarrow(z^s \mid C)$ describes the coarse-scale distribution induced at scale $s$ by deterministic temporal downsampling when the finest-scale latent variable follows $p(z^S \mid C)$.

Based on this derivation, cross-scale probabilistic consistency requires the directly modeled distribution at scale $s$ to be compatible with the downsampling-induced distribution:

$$p(z^s \mid C) \approx p_\downarrow(z^s \mid C). \tag{24}$$

This condition indicates that latent variables at different scales should represent the same underlying action trajectory at different temporal resolutions, rather than independent or even conflicting behavior distributions. For MSP, this consistency is not enforced by introducing an additional explicit distribution-matching loss. Instead, it is naturally induced by the model design. Specifically, the latent targets at all scales are constructed from the same finest-scale latent action trajectory through deterministic temporal downsampling, so they inherently share the same underlying action semantics in the training objective. During generation, MSP further adopts coarse-to-fine autoregressive conditional modeling, where fine-scale latent variables are generated conditioned on previously produced coarse-scale semantics, progressively refining global action intent into local temporal details.

Therefore, MSP promotes cross-scale probabilistic consistency through multi-scale latent construction and the coarse-to-fine generation process, rather than relying on an additional explicit divergence constraint. This design maintains probabilistic compatibility across different temporal scales while avoiding the training complexity and instability that may arise from explicit cross-scale distribution matching.

## B. Experimental and Dataset

We conduct a series of systematic experiments, including two simulation benchmarks comprising 138 tasks in total and real-world experiments spanning five different temporal horizon levels. These experiments involve multi-modal inputs, including 2D images, 3D point clouds, and language embeddings.

### B.1. Simulated Environments

**LIBERO.** LIBERO is a lifelong learning benchmark consisting of five task suites: LIBERO-Goal, LIBERO-Spatial, LIBERO-Object, and LIBERO-Long, each containing 10 tasks with 50 expert demonstrations, as well as LIBERO-90, which includes 90 tasks for large-scale multi-task evaluation, resulting in a total of 130 tasks. All tasks are performed using a Franka Panda arm with a 7-dimensional action space and 9-dimensional pose observations, with 2D images used as visual inputs. In this work, we use LIBERO-90 to study large-scale multi-task learning, LIBERO-Long to evaluate long-horizon scenarios, and LIBERO-90 for pretraining a policy that is subsequently fine-tuned on LIBERO-Long to assess few-shot generalization. The remaining task suites are used to evaluate short-horizon performance and multi-scene adaptability.

For each task, we train the model using 50 expert demonstrations. During evaluation, each task is executed 50 times using the best checkpoints from three different random seeds, and we report the average success rate for each suite.

**RoboTwin2.0.** RoboTwin2.0 is a bimanual robotic manipulation benchmark designed to capture real-world complexity and enable robust sim-to-real transfer. It comprises 50 tasks with varying temporal horizons, spans multiple robot embodiments, and includes 731 object instances. For training and evaluation on RoboTwin2.0, we use an Agilex Piper robotic arm with a 14-dimensional action space, and adopt 2D images and 3D point clouds as observational inputs, allowing arbitrary numbers of demonstrations to be generated. For systematic evaluation, we select eight representative tasks based on their average execution lengths and group them into four temporal horizon levels.

We train each task using 50 clean expert demonstrations, and evaluate performance on 100 independent held-out test scenarios per task.

## C. Implementation Details

The main contribution of this work lies in action generation design; therefore, the visual backbones of all comparison methods are kept consistent. For the LIBERO simulation benchmark, following LBP, we filter out failed trajectories in the dataset and resize all images to $224 \times 224$. We adopt the backbone settings from LBP, using a ResNet-34 backbone from DecisionNCE (Li et al., 2024) to extract visual features from different camera views, with language embeddings injected via FiLM conditioning layers. For the RoboTwin2.0 simulation benchmark, both 2D image and 3D point cloud implementations follow the same visual processing pipelines as DP and DP3. All tasks are trained on two TITAN XP servers and evaluated on a single RTX 4060 Ti (16GB). Inference speed is measured on an RTX 4060 Ti GPU by averaging the runtime of 400 action predictions (excluding the visual backbone) over five runs to ensure robustness. See Table 5 for the efficiency comparison. Detailed training hyperparameters are provided in Table 6.

*Table 5.* **Comparison of Parameters, Inference Time, and Prediction Horizon.** We compare various methods based on their model size, inference speed, and prediction steps.

| Method | Params/M | Inference Time/ms | Prediction Horizon/step |
|---|---|---|---|
| DP-C | 262.35 | 88.53 | 16 |
| FlowPolicy | 259.32 | 12.63 | 4 |
| CARP | 16.08 | 40.87 | 16 |
| DSP | 28.77 | **5.05** | 16 |
| MSP | **17.2** | 18.66 | **32** |

*Table 6.* **Hyperparameter Settings for MSP.** Task Num: number of tasks in the benchmark; ObsRes: environment observation resolution (camera views $\times W \times H$; point cloud views $\times N \times C$); To: observation horizon; Ta: action horizon; TE: whether to apply a Temporal Ensemble strategy to the executed actions, as in ACT; if not, the entire action chunk is executed in a single inference; Tp: action prediction horizon; Stage1-Bs: batch size in Stage 1 (continuous latent action modeling); Stage1-Es: number of training epochs in Stage 1; Stage2-Bs: batch size in Stage 2 (probabilistically consistent multi-scale modeling); Stage2-Es: number of training epochs in Stage 2; Lr: learning rate; Optimizer: training optimizer.

| H-Param | Task Num | ObsRes | To | Ta | TE | Tp | Stage1–Bs | Stage1–Es | Stage2–Bs | Stage2–Es | Lr | Optimizer |
|---|---|---|---|---|---|---|---|---|---|---|---|---|
| LIBERO_Long | 10 | 2x224x224 | 1 | 8 | Yes | 32 | 512 | 400 | 64 | 200 | 3e-4 | AdamW |
| LIBERO_90 | 90 | 2x224x224 | 1 | 8 | Yes | 32 | 512 | 400 | 64 | 200 | 3e-4 | AdamW |
| LIBERO_Object | 10 | 2x224x224 | 1 | 8 | Yes | 32 | 256 | 600 | 64 | 200 | 3e-4 | AdamW |
| LIBERO_Spatial | 10 | 2x224x224 | 1 | 8 | Yes | 32 | 256 | 600 | 64 | 200 | 3e-4 | AdamW |
| LIBERO_Goal | 10 | 2x224x224 | 1 | 8 | Yes | 32 | 256 | 600 | 64 | 200 | 3e-4 | AdamW |
| RoboTwin2.0 (2D) | 8 | 1x320x240 | 2 | 32 | No | 32 | 256 | 600 | 32 | 300 | 3e-4 | AdamW |
| RoboTwin2.0 (3D) | 8 | 1x1024x6 | 2 | 32 | No | 32 | 256 | 600 | 32 | 300 | 3e-4 | AdamW |
| Real–World | 5 | 3x320x240 | 2 | 32 | No | 32 | 256 | 600 | 32 | 300 | 3e-4 | AdamW |

## D. Real-Robot Experiment Details

To validate the effectiveness of MSP in real-world settings, we design five tasks with different temporal horizons:

1. *Pick apple*: grasp an apple and place it onto a plate;

2. *Hang cup*: grasp the rim of a cup, rotate it to adjust its orientation, and hang the handle onto a cup holder;

3. *Stack cups*: press a timer to start, stack the first cup into the second, then grasp the two stacked cups and place them into a third cup, and finally press the timer to end;

4. *Place object cabinet*: pull out the cabinet drawer, adjust the grasping angle to pick up a banana and place it into the cabinet, then adjust the grasp to pick up a rectangular wooden block and place it inside, and finally close the drawer by pushing;

5. *Make tea*: take a teapot from the shelf, open the lid, place a tea bag into the teapot, take a cup from the lower shelf, and pour tea into the cup.

For our stage-based scoring protocol, we use the *Pick apple* task as an example: approaching the apple is scored as 25 points, successfully grasping the apple as 50 points, moving it above the plate as 75 points, and finally placing it onto the plate receives the full score of 100 points. For multi-stage tasks, the score of the next stage is counted only after the current stage achieves a full score. Table 7 reports detailed success rates for each sub-stage of all tasks.

*Table 7.* **Average Score across Multi-Stage Tasks.** We compare MSP against DSP and DP across 5 tasks with varying horizon stages.

| Task | Stage | Average Score | | |
|---|---|---|---|---|
| | | MSP | DSP | DP |
| Pick apple | Stage 1 | 91.67 | 85.00 | 78.30 |
| Hang cup | Stage 1 | 95.00 | 86.67 | 80.83 |
| | Stage 2 | 78.30 | 60.00 | 47.50 |
| Stack cups | Stage 1 | 90.00 | 80.00 | 85.00 |
| | Stage 2 | 75.83 | 61.67 | 46.67 |
| | Stage 3 | 60.00 | 41.67 | 28.33 |
| Put object cabinet | Stage 1 | 95.83 | 91.67 | 84.17 |
| | Stage 2 | 81.67 | 63.33 | 48.33 |
| | Stage 3 | 69.17 | 45.00 | 21.67 |
| | Stage 4 | 48.33 | 30.00 | 3.33 |
| Make tea | Stage 1 | 90.83 | 85.00 | 79.17 |
| | Stage 2 | 75.83 | 58.33 | 42.50 |
| | Stage 3 | 65.00 | 41.67 | 15.83 |
| | Stage 4 | 47.50 | 13.33 | 0.00 |
| | Stage 5 | 28.33 | 0.00 | 0.00 |

# E. Additional Benchmark Comparisons

$H^3DP$ is one of the most relevant baselines to MSP, as it also adopts a multi-scale formulation in continuous action spaces. Since the code of $H^3DP$ is not publicly available, we evaluate MSP directly on the RoboTwin1.0 (Mu et al., 2025) benchmark adopted by $H^3DP$, avoiding potential bias from unofficial reimplementation. Under fully aligned settings, including identical camera views and expert demonstrations, MSP achieves an average success rate of 69.63% across 8 tasks, outperforming $H^3DP$ by +12.25%, as shown in Table 8.

*Table 8.* **Performance Comparison between $H^3DP$ and MSP on RoboTwin1.0.** We compare MSP against $H^3DP$ across 8 RoboTwin1.0 tasks.

| Task | $H^3DP$ (%) | MSP (%) |
|---|---|---|
| Apple Cabinet Storage | 98 | **100** |
| Dual Bottles Pick (Easy) | 48 | **93** |
| Dual Bottles Pick (Hard) | **53** | 51 |
| Block Handover | 70 | **95** |
| Block Hammer Beat | **85** | 81 |
| Diverse Bottles Pick | 25 | **51** |
| Pick Apple Messy | **35** | 27 |
| Tool Adjust | 45 | **59** |
| **Average** | 57.38 | **69.63** |

To further assess the generality of MSP, we extend the evaluation to 8 additional RoboTwin2.0 tasks with diverse temporal horizons, ranging from 85 to 537 steps. We compare MSP3D with the DP3 baseline, and report the task sequence lengths and success rates in Table 9. The results show that MSP consistently achieves stronger performance across diverse manipulation scenarios, with particularly clear gains on longer-horizon tasks.

*Table 9.* **Performance Comparison between DP3 and MSP3D on Additional RoboTwin2.0 Tasks.** We compare MSP3D against DP3 across 8 additional RoboTwin2.0 tasks with different trajectory lengths.

| Task (Steps) | DP3 (%) | MSP3D (%) |
|---|---|---|
| click alarmclock (85) | 77 | **98** |
| beat block hammer (113) | **72** | 70 |
| place bread basket (231) | 26 | **38** |
| open laptop (258) | **82** | 80 |
| dump bin bigbin (265) | 85 | **93** |
| place cans plasticbox (289) | 48 | **91** |
| hanging mug (340) | 17 | **27** |
| open microwave (537) | 61 | **99** |
| **Average** | 58.5 | **74.5** |

In addition, we evaluate MSP on the Push-T simulation benchmark (Chi et al., 2023). Push-T provides a rigorous testbed for evaluating a model's ability to capture complex contact dynamics and multimodal action distributions. Under identical experimental conditions, MSP achieves a success rate of 94.2%, outperforming the strong DP-C baseline by +3.2%, as shown in Table 10.

*Table 10.* **Performance on the Push-T Benchmark.** We compare MSP with representative baselines on the Push-T simulation benchmark.

| Method | Success Rates (%) |
|---|---|
| DP-C | 91.0 |
| DP-T | 78.8 |
| ACT | 77.5 |
| OpenVLA | 35.0 |
| **MSP** | **94.2** |

# F. Training and Inference Pipeline

Algorithms 1 and 2 summarize the training and inference procedures of MSP. During training, we first learn a smooth continuous latent action space with a VAE, and then freeze it to train the scale-wise autoregressive Transformer and the CFG-aware MeanFlow module under a unified scale-wise flow matching objective. The finest latent representation is downsampled into a hierarchy of coarse-to-fine latents. Conditioned on the observation embedding and previous coarse-scale latents, the Transformer predicts semantic conditions for each scale, while the scale-wise causal attention mask prevents access to current or finer-scale ground-truth latents, thereby avoiding information leakage. The MeanFlow module is then optimized at each scale using the Transformer-generated semantic condition, which constrains it to learn structured hierarchical dependencies rather than shortcut mappings. In addition, the CFG-aware formulation and scale-dependent loss weights further stabilize training by balancing conditional distribution modeling across different temporal resolutions. During inference, MSP recursively generates latents from coarse to fine, where each scale is sampled from Gaussian noise with one MeanFlow function evaluation and then upsampled to condition the next scale. The final latent $z^S$ is decoded into the executable action sequence.

---

**Algorithm 1** Training of MSP

---

1: **Inputs:** Demonstration dataset $\mathcal{D} = \{\tau_i\}_{i=1}^N$, where $\tau = \{(o_t, a_t)\}_{t=1}^H$
2: **Hyperparameters:** Action horizon $T$, latent length $T'$, number of scales $S$, downsampling factors $\{r_s\}_{s=1}^S$, where $r_s = 2^{S-s}$ and $t_s = T'/r_s$
3: **Initialize:** VAE encoder $E_\phi$, VAE decoder $D_\phi$, observation encoder $f_\psi$, scale-wise Transformer $\mathcal{T}_\theta$, CFG-aware MeanFlow model $u_\theta^{\text{cfg}}$
4: **Stage 1: Continuous latent action modeling**
5: **for** each training iteration **do**
6:     Sample $(o_t, a_{t:t+T-1}) \sim \mathcal{D}$
7:     $q_\phi(z \mid a_{t:t+T-1}) \leftarrow E_\phi(a_{t:t+T-1}), z \sim q_\phi(z \mid a_{t:t+T-1}), \hat{a}_{t:t+T-1} \leftarrow D_\phi(z)$
8:     $\mathcal{L}_{\text{act}} \leftarrow \|\hat{a}_{t:t+T-1} - a_{t:t+T-1}\|_1 + \beta D_{\text{KL}}\big(q_\phi(z \mid a_{t:t+T-1})\|\mathcal{N}(0, I)\big)$
9:     Update $\phi$ by minimizing $\mathcal{L}_{\text{act}}$
10: **end for**
11: **Stage 2: Probabilistically consistent multi-scale latent generation**
12: Freeze $E_\phi$ and $D_\phi$
13: **for** each training iteration **do**
14:     Sample $(o_t, a_{t:t+T-1}) \sim \mathcal{D}$ and compute $q_\phi(z \mid a_{t:t+T-1}) \leftarrow E_\phi(a_{t:t+T-1}), z \sim q_\phi(z \mid a_{t:t+T-1})$
15:     $C \leftarrow f_\psi(o_t)$
16:     Construct $\mathcal{Z} = \{z^1, z^2, \ldots, z^S\}$, where $z^s = \text{Down}(z, r_s)$
17:     $\hat{Z} = \{\hat{z}^1, \ldots, \hat{z}^S\} \leftarrow \mathcal{T}_\theta\big([C, \text{Up}(z^1, 2), \ldots, \text{Up}(z^{S-1}, 2)]\big)$
18:     $\mathcal{L}_{\text{MSP}} \leftarrow 0$
19:     **for** $s = 1$ to $S$ **do**
20:         Sample $\epsilon \sim \mathcal{N}(0, I)$ and time interval $(r, t)$; set $x_t^s \leftarrow (1-t)z^s + t\epsilon$
21:         Compute the MeanFlow target $u_{\text{tgt}}^s$
22:         $\mathcal{L}_{\text{cfg}}^{(s)} \leftarrow \big\|u_\theta^{\text{cfg}}(x_t^s, r, t \mid \hat{z}^s) - \text{sg}(u_{\text{tgt}}^s)\big\|_2^2$
23:         $\mathcal{L}_{\text{MSP}} \leftarrow \mathcal{L}_{\text{MSP}} + \frac{s}{\max(Z)}\mathcal{L}_{\text{cfg}}^{(s)}$
24:     **end for**
25:     Update $f_\psi$, $\mathcal{T}_\theta$, and $u_\theta^{\text{cfg}}$ by minimizing $\mathcal{L}_{\text{MSP}}$
26: **end for**

---

---

**Algorithm 2** Inference of MSP

---

1: **Inputs:** Current observation $o_t$
2: **Hyperparameters:** Action horizon $T$, latent length $T'$, number of scales $S$, downsampling factors $\{r_s\}_{s=1}^S$, where $r_s = 2^{S-s}$ and $t_s = T'/r_s$
3: **Load trained modules:** Observation encoder $f_\psi$, scale-wise Transformer $\mathcal{T}_\theta$, CFG-aware MeanFlow model $u_\theta^{\text{cfg}}$, VAE decoder $D_\phi$
4: $C \leftarrow f_\psi(o_t), X \leftarrow C$
5: **for** $s = 1$ to $S$ **do**
6:     $\hat{z}^s \leftarrow \mathcal{T}_\theta(X)$
7:     Sample $\epsilon^s \sim \mathcal{N}(0, I)$ with length $t_s$
8:     $r \leftarrow 0, t \leftarrow 1, x_t^s \leftarrow \epsilon^s$
9:     $z^s \leftarrow x_t^s - (t-r)u_\theta^{\text{cfg}}(x_t^s, r, t \mid \hat{z}^s)$
10:     **if** $s < S$ **then**
11:         $X \leftarrow \text{Up}(z^s, 2)$
12:     **end if**
13: **end for**
14: $\hat{a}_{t:t+T-1} \leftarrow D_\phi(z^S)$
15: **Return:** Executable action sequence $\hat{a}_{t:t+T-1}$

---

## G. More Visualization Results

Additional visualizations of MSP rollouts are provided in Figure 9 to 11. For the LIBERO benchmark, we visualize one representative task from each suite, while for RoboTwin2.0, we showcase challenging long-horizon tasks. Furthermore, we illustrate real-robot execution performance from our real-world experiments.

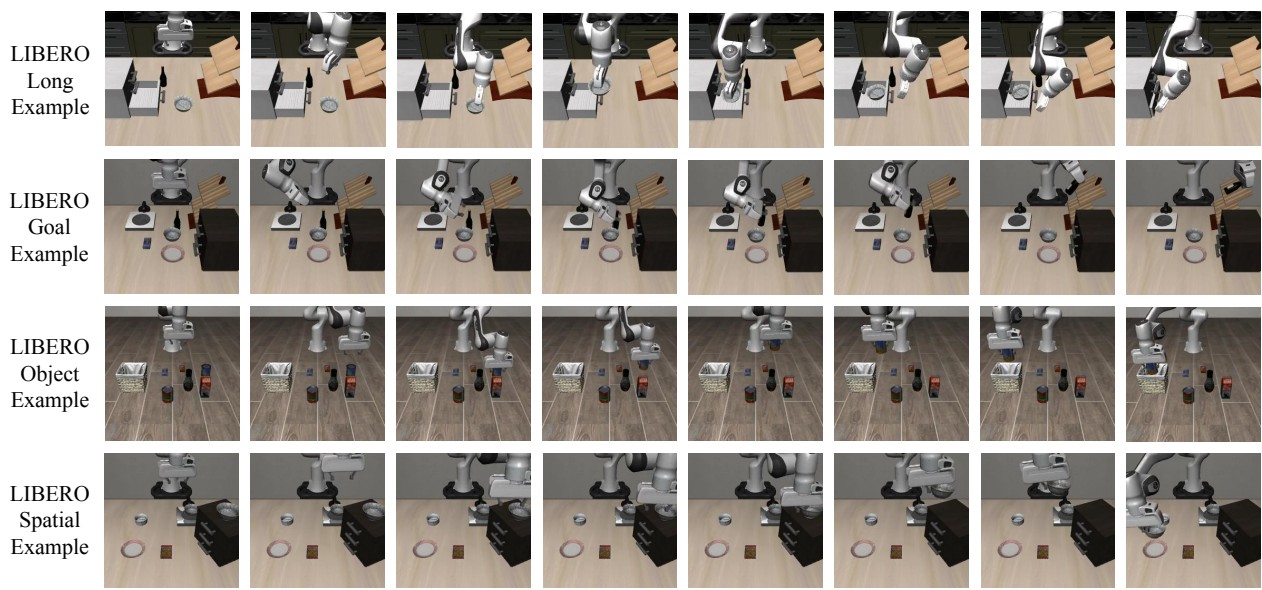

*Figure 9.* **Visualization of LIBERO Task Experiments**

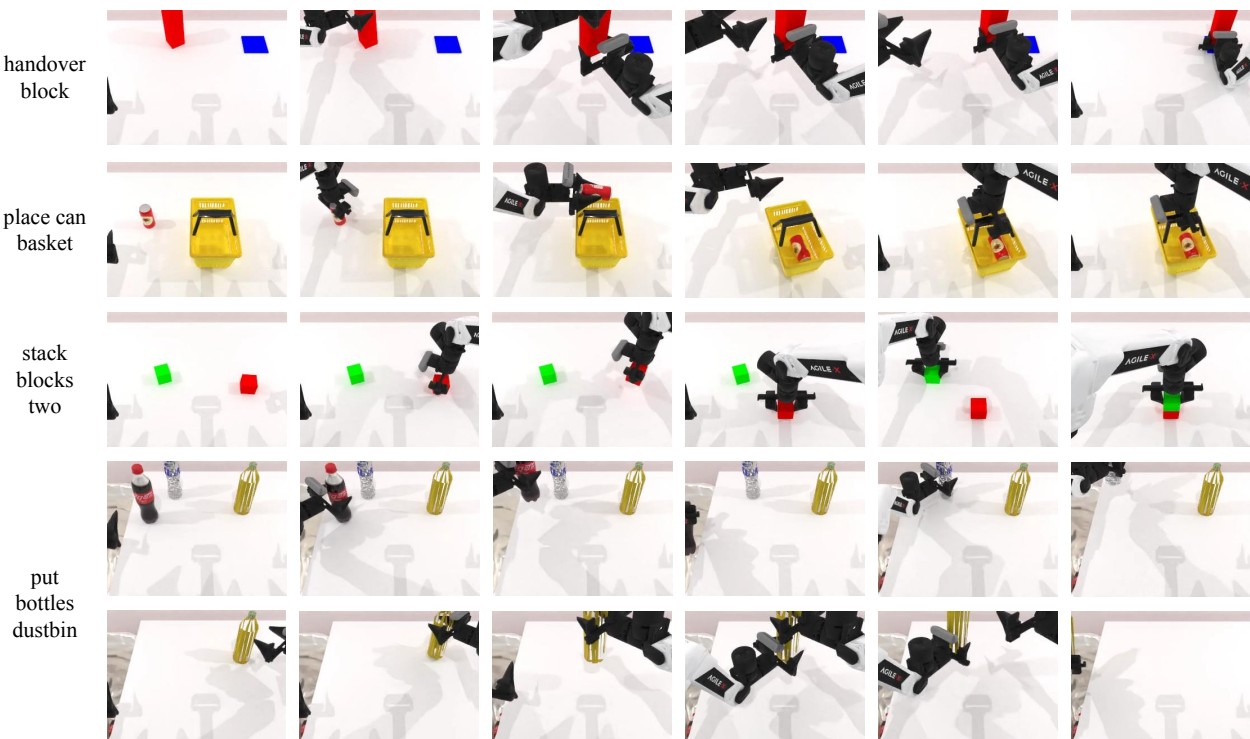

*Figure 10.* **Visualization of RoboTwin2.0 Task Experiments**

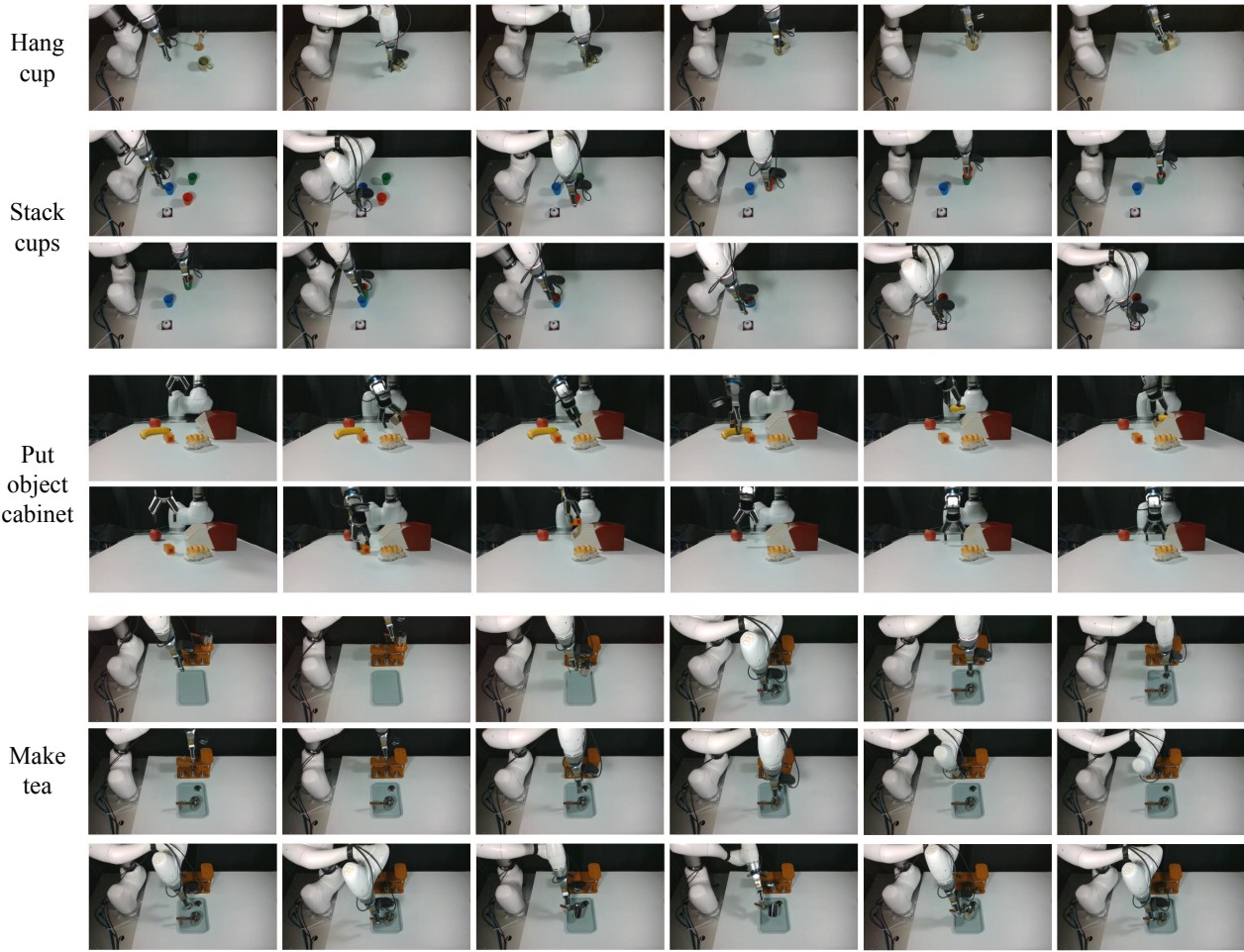

*Figure 11.* **Visualization of Read-Word Task Experiments**

