# OpenReview forum: "MSP: Probabilistically Consistent Multi-Scale Action Generation"
_ICML.cc/2026/Conference — ICML 2026 spotlight_

### Official Review · Reviewer_BW4Z · 2026-03-11

**Soundness:** 3
**Presentation:** 2
**Significance:** 3
**Originality:** 3
**Overall Recommendation:** 4
**Confidence:** 4

**Summary:**

This paper proposes a Probabilistically Consistent Multi-Scale Action Generation (MSP) framework for coarse-to-fine action generation. The method first compresses action sequences using a VAE to obtain multi-scale representations, and then employs a scale-wise autoregressive Transformer to progressively generate semantic conditions across different temporal scales. This design aims to maintain probabilistic consistency across scales while enabling hierarchical action generation. Experimental results demonstrate promising performance compared with existing baselines.

**Compliance With Llm Reviewing Policy:**

Affirmed.

**Final Justification:**

I think the rebuttal has addressed my concern.

**Key Questions For Authors:**

1) What exactly does “probabilistic consistency” mean in this work? A formal definition or intuitive explanation would help clarify the main idea of the paper.

2)  Why does the scale-wise autoregressive module perform downsampling followed by upsampling? What advantages does this design provide compared to directly modeling sequences at different resolutions?

3) What are the key differences between this method and FlowAR (Ren et al., 2025)?
Since the paper explicitly cites FlowAR as inspiration, it would be helpful to clearly articulate the novelty beyond that work.

4) What does the variable d represent in Line 180?

5) Why do autoregressive policies in continuous action spaces fail to preserve hierarchical semantic coherence?
Can the authors provide theoretical intuition or empirical evidence supporting this claim? Additionally, how do the experiments demonstrate that the proposed method addresses this limitation?

I vote for weak reject at this moment because the paper has some many unclear points. Happy to upgrade if others are more positive and if authors can address the concerns.

**Limitations:**

The main motivation is "Sequential actions are often highly correlated, making it crucial to model the skill semantics and temporal relationships in long action sequences.", however, there can also be some unrelated activities in a sequence due to completing different tasks, like eating, then maybe take a rest, and do other unrelated tasks.

**Strengths And Weaknesses:**

Strengths:

1) Well-motivated problem. The paper studies hierarchical action generation and proposes a coarse-to-fine framework to improve semantic consistency across temporal scales. The motivation is reasonable and the task itself is important for sequential decision-making and generative modeling.

2) Potentially interesting modeling design. The proposed Probabilistically Consistent Multi-Scale (MSP) framework combines VAE-based compression with a scale-wise autoregressive Transformer. This hierarchical generation strategy appears conceptually interesting and could be useful for modeling structured action sequences.

3) Potential originality. The idea of enforcing probabilistic consistency across multiple temporal scales appears relatively novel, at least in the context of action generation.

4) Clear visual presentation. The figures are well-designed and help convey the overall pipeline. However, the text does not always sufficiently reference or explain the figures, which limits their usefulness.

Weaknesses:

1) Key concepts are insufficiently defined. The notion of probabilistic consistency appears central to the paper, but the paper does not clearly define what this term means in practice.  A formal definition or clearer intuition would significantly improve readability.

2) Insufficient justification of the hierarchical generation design. The scale-wise autoregressive module involves downsampling actions and then upsampling them during generation. The paper does not clearly explain the motivation behind this design choice. In particular, it would be helpful to clarify: a. why the downsampling–upsampling process is necessary. b. what advantages it provides over directly modeling the full-resolution sequence.  c. whether this design improves efficiency, abstraction, or learning stability.

3) Unclear novelty relative to FlowAR. The paper mentions that the approach is inspired by FlowAR (Ren et al., 2025) and extends it to a multi-scale framework with coarse-to-fine latent sequence generation. However, the paper does not clearly explain: a. the key conceptual or technical differences between the proposed method and FlowAR. b. what new components are introduced beyond FlowAR,
or why the extension is non-trivial. Additionally, FlowAR is not sufficiently discussed in the related work section, making it difficult to assess the novelty of the contribution.

4) Missing or unclear technical details. Some symbols and variables are introduced without proper explanation. For example, the variable d in Line 180 is not defined clearly. Such missing details make it difficult to fully understand the model specification.

5) The central motivation is not clearly validated experimentally. The paper claims that autoregressive policies in continuous action spaces struggle to preserve coherent semantics across hierarchical levels, which motivates the proposed coarse-to-fine framework. However: the paper does not clearly explain why this limitation arises,  it is unclear whether prior work has demonstrated this issue, and the experiments do not convincingly show that the proposed method specifically resolves this problem. Since this claim appears to be a key motivation for the method, stronger empirical evidence or analysis would strengthen the paper.

6) Weak integration between figures and the main text. Although the figures are visually clear, the text does not sufficiently reference or explain them in detail. As a result, readers may find it difficult to connect the visual diagrams with the technical descriptions of the method

---

> ### Author Rebuttal · Authors · 2026-03-30
>
> Dear Reviewer,
>
> Thank you very much for your constructive feedback and insightful comments. Due to strict character limits, we have summarized your questions as Q1–Q5 to prioritize a comprehensive response. Our answers correspond to your points in order:
>
> ***Q1: Definition of Probabilistic Consistency.***
>
> **A1:** Thank you for highlighting the need to clarify the definition and implementation of probabilistic consistency. In the MSP framework, probabilistic consistency ensures that fine-grained local control remains strictly within the probabilistic boundaries defined by coarse-grained intentions in the continuous latent action space. Formally, we define this property as *the downsampling equivalence of the cross-scale distribution*, requiring the directly generated coarse-scale distribution to match the marginal distribution induced from the highest-resolution latent action space:
>
> $$p(z^s | C) \equiv \int p(z^S | C) \cdot \delta(z^s - \text{Down}(z^S, r_s)) \ dz^S$$
>
> Here, the left-hand side represents the conditional distribution at scale $s$ directly predicted under the observation condition $C$. The right-hand side represents the total probability obtained by compressing the highest-resolution sequence $z^S$ to resolution $r_s$ via $\text{Down}(z^S, r_s)$, followed by marginalization over the highest-resolution space using the Dirac delta function $\delta(\cdot)$. This equivalence ensures that fine-grained local control is probabilistically consistent with coarse-scale initial semantics.
>
> During training, Eq. (11) optimizes this objective: scale-wise MeanFlow loss forces the model to approximate the target distribution that naturally satisfies this consistency. Furthermore, scale-based weights ensure high-fidelity alignment with semantic conditions $\hat{z}^s$, while the scale-wise MeanFlow model's 1-NFE generation avoids the iterative integration process, maintaining cross-scale probabilistic consistency. In short, the equation defines the ideal target, while Eq. (11) provides the optimization path.
>
> ***Q2: Rationality of the Downsample-Upsample Architecture.***
>
> **A2:** In the MSP framework, downsampling constructs a multi-scale hierarchy in the latent action space, defining the target latents at each scale, while upsampling enables recursive refinement. During inference, latent actions are generated scale-wise in an autoregressive manner. Once the sequence at scale $s-1$ is produced, $Up(z,2)$ expands its temporal dimension to match scale $s$, and the result is fed into the Transformer to produce the semantic condition $\hat{z}^s$.
>
> To maintain training–inference consistency, the observation condition serves as the initial scale input, and higher scales during training are constructed by upsampling downsampled ground-truth latents. This encourages the model to learn consistent transitions between scales and preserves a coherent cross-scale trajectory.
>
> Direct modeling of long, high-dimensional sequences often overfits local details and weakens global structure. In contrast, our design follows a coarse-to-fine planning principle, mapping global intentions to local execution while preserving probabilistic consistency across scales. As shown in Table 3, direct modeling without hierarchical constraints achieves 88.4% success, whereas the full multi-scale architecture reaches 96.8%, demonstrating the effectiveness of the hierarchical design.
>
> ***Q3: Novelty beyond FlowAR.***
>
> **A3:** While inspired by FlowAR, MSP introduces several non-trivial extensions for real-time continuous robot control. First, whereas FlowAR targets 2D image generation, MSP adapts the framework to continuous latent action sequences representing robot trajectories. To achieve this, we utilize a VAE framework to balance latent smoothness and generation diversity. Second, standard flow-matching in FlowAR relies on an iterative integration process, which is inefficient for real-time control. To address this, we integrate a lightweight scale-wise MeanFlow model, enabling accurate 1-NFE generation at each scale and supporting efficient real-time execution.
>
> ***Q4: Clarification of Variable $d$.***
>
> **A4:** In the expression $z \in \mathbb{R}^{t_s \times d}$, the variable $d$ denotes the feature dimension of the continuous latent action representation at a given downsampled time step. We will explicitly define this in the revised manuscript.
>
> ***Q5: Addressing Failure in Standard AR Policies.***
>
> **A5:** Standard continuous autoregressive (AR) policies operate in a step-wise manner, where minor high-frequency prediction errors accumulate across long sequences. Lacking an explicit hierarchy, these local deviations cause the policy to rapidly drift off-distribution. Our experiments substantiate this limitation: while the baseline DSP degrades significantly on extra-long horizon tasks, MSP outperforms DSP by 6.8% on LIBERO-Long, demonstrating the superiority of our multi-scale approach in ensuring long-term coherence.

---

> > ### Author Rebuttal · Reviewer_BW4Z · 2026-04-01
> >
> > Thanks for the response. my concerns are  all addressed now.

---

> > > ### Author Response · Authors · 2026-04-03
> > >
> > > Dear Reviewer,
> > >
> > > We deeply appreciate the time and energy you dedicated to reviewing our work! It is wonderful to hear that our detailed explanations have successfully resolved your concerns.
> > >
> > > We hope you are having a good week.

---

### Official Review · Reviewer_DGyV · 2026-03-13

**Soundness:** 3
**Presentation:** 3
**Significance:** 2
**Originality:** 2
**Overall Recommendation:** 4
**Confidence:** 4

**Summary:**

This paper tackles the challenge of long-horizon imitation learning for robotic manipulation, where global task intent must coexist with precise local control over extended action sequences. Existing coarse-to-fine methods either rely on discrete codebooks or operate in raw continuous action spaces without explicit cross-scale structure, causing compounding errors over long horizons. The authors propose MSP (Probabilistically Consistent Multi-Scale Action Generation), a two-stage framework. In Stage 1, a causal Transformer within a VAE encodes raw action sequences into a smooth, temporally structured continuous latent space. In Stage 2, this latent is downsampled to construct a multi-scale representation $Z$. A scale-wise autoregressive Transformer generates conditional semantic features $\hat{z}^s$ for each scale conditioned on coarser scales, and a lightweight scale-wise MeanFlow model maps noise to the target latent at each scale using a single function evaluation (1-NFE). Classifier-Free Guidance is incorporated throughout. At inference, MSP recursively generates latents from coarse to fine, decoding only the finest scale into executable actions. Experiments span LIBERO, RoboTwin2.0, multi-task generalization on LIBERO-90, few-shot transfer with 5 demonstrations, and real-world deployment on a Doosan arm with 5 progressively longer tasks. MSP consistently outperforms strong baselines, achieving an average 96.9% on LIBERO and 82.25% on RoboTwin2.0 under 3D inputs, with advantages growing with horizon length.

**Compliance With Llm Reviewing Policy:**

Affirmed.

**Final Justification:**

The rebuttal addresses all of my concerns.

**Key Questions For Authors:**

1. The paper's central claim is that MSP achieves probabilistic consistency across scales, distinguishing it from prior work. Please provide a formal mathematical definition of what this consistency property is, what its violation would imply for downstream task performance, and how the factored distribution in Eq. (3) constitutes a stronger or more meaningful consistency guarantee than prior methods.

2. H3DP appears to be the most directly comparable multi-scale, continuous-action-space baseline. Why is it absent from Tables 1 and 2? If it is unavailable for comparison, what qualitative differences in architecture or methodology does MSP offer over H3DP that would predict the performance gap?

3. The MeanFlow Identity (Eq. 6–7) is applied independently at each of $S=4$ scales. In Stage 2, the semantic conditioning $\hat{z}^s$ for scale $s$ depends on the generated latent $z^{s-1}$ from the previous scale. If the MeanFlow model at scale $s-1$ produces an inexact $z^{s-1}$ (due to approximation in the average velocity field), this error propagates to $\hat{z}^s$ and thence to $z^s$. Does this cross-scale error propagation compound across scales? Have the authors measured the individual per-scale reconstruction accuracy, and does the stop-gradient in $sg(u_{tgt})$ prevent gradient-level error amplification during training?

4. The training objective (Eq. 11) weights scale $s$ by $\frac{s}{max(Z)}$, giving the finest scale $(s=4)$ $4×$ the weight of the coarsest $(s=1)$. Why is this the correct weighting? The coarsest scale provides the semantic anchor for all subsequent refinements, shouldn't its accuracy be prioritized? Please provide an ablation comparing uniform weighting, inverse weighting (coarser scales weighted more), and the proposed scheme.

5. Stage 1 trains the VAE independently of Stage 2. If the VAE cannot accurately reconstruct certain motion patterns (e.g., high-frequency contact dynamics), Stage 2 cannot recover this information from the latent. What is the Stage 1 reconstruction quality (mean absolute error in action space) for each benchmark, and how does this affect the theoretical performance ceiling? Have the authors considered end-to-end fine-tuning of Stage 1 and Stage 2 jointly, and if so, does it improve performance?

**Limitations:**

The paper includes an impact statement but an explicit limitations section is absent, and should be added in the next version.

**Strengths And Weaknesses:**

**Strengths:**
- The two-stage design is methodologically clean and well-motivated. Stage 1 VAE provides a smooth, differentiable latent space that avoids the quantization artifacts of codebook-based methods, while Stage 2 explicitly models the multi-scale joint distribution via autoregressive factorization (Eq. 3).
- The integration of MeanFlow (Geng et al., 2025) into the scale-wise generation is a well-chosen design: by learning the average velocity field rather than the instantaneous one, 1-NFE inference is achieved naturally without distillation. The MeanFlow Identity (Eq. 6–7) and its use of a stop-gradient $sg(\cdot)$ to avoid double backpropagation through Jacobian-vector products is technically sound and practically important.
- The ablation studies (Table 3) are systematic and genuinely informative: removing each of the four core components (VAE latent, multi-scale structure, diffusion vs. flow matching vs. MeanFlow) leads to distinct and measurable performance drops (96.8% → 89.6% → 82.2% → 79.6% → 79.2%), confirming that each element is necessary.
- The real-world evaluation (Table 6 / Figure 8) is thorough and shows consistent advantages that compound across stages, e.g., Make tea Stage 5: MSP 28.33 vs. DSP 0.00 and DP 0.00, providing strong evidence that cross-scale consistency directly reduces error accumulation in deployed multi-stage tasks.

**Weaknesses:**
- The central concept of "probabilistic consistency" is never formally defined. Despite appearing in the title, abstract, and throughout the text as the core contribution, the paper never states a mathematical definition of what that means or what it would mean to violate it. The factored joint distribution $p(z^{1:S}) = \Pi_s p(z^s | z^{<s}, \theta)$ in Eq. (3) is simply the standard chain rule decomposition, this is not a novel consistency constraint. A formal definition, e.g., "the marginal distribution at scale s is consistent with the marginal induced by the fine-scale distribution" would ground the claim and allow falsification.

- The scale loss weighting $\frac{s}{max(Z)}$ in Eq. (11) is an unjustified design choice. This weighting assigns 4× more weight to the finest scale than the coarsest. While the authors state this aligns the contribution of each scale with its latent size, this is not a principled argument. The coarsest scale governs the global semantic structure that all finer scales depend on; it is not obvious that it should receive the least learning signal. No ablation on alternative weighting schemes (e.g., uniform, inverse, or learned weights) is provided.

- The MeanFlow Identity (Eq. 6) is adopted without derivation. The key mathematical transformation from the integral definition of the average velocity field $u$ (Eq. 5) to the locally differentiable identity (Eq. 6) is stated with a reference to Geng et al. (2025) but not derived or even sketched in the paper. In a multi-scale hierarchical setting, this identity is applied $S=4$ times with potentially correlated prediction errors across scales. Whether the approximation errors in the supervised signal $u_{tgt}$ (Eq. 7) compound across scales and whether the stop-gradient prevents this is not analyzed. The paper should at minimum provide the one-line derivation of the MeanFlow Identity and address this potential cross-scale error accumulation.

- The VAE reconstruction quality is unreported. Stage 1 establishes the latent ceiling for all subsequent generation: if the VAE cannot faithfully reconstruct action sequences, MSP's success rate is upper-bounded by reconstruction quality. No reconstruction error, sequence-level success rate of the decoder, or latent visualization is provided to characterize this ceiling.

- H3DP (Lu et al., 2025) is cited in the related work as a hierarchical diffusion policy that tightly couples perceptual features with coarse-to-fine action generation, the most structurally similar baseline, but is absent from Tables 1 and 2. Given that H3DP and MSP share the multi-scale + flow-based generation paradigm in continuous action spaces, their comparison is essential to establish MSP's novelty advantage.

- The core architectural components: VAE for continuous latent action spaces, autoregressive transformer for coarse-to-fine conditioning, and MeanFlow for 1-NFE generation, are all pre-existing building blocks from recent work (MP1, FlowAR, MeanFlow). The paper's primary contribution is their combination and the multi-scale latent downsampling design, which is incremental rather than conceptually novel.

---

> ### Author Rebuttal · Authors · 2026-03-30
>
> Dear Reviewer,
>
> We sincerely thank you for the constructive feedback. Due to strict character limits, we summarize your questions as Q1–Q5. To ensure comprehensive answers within space constraints, we kindly cross-reference overlapping questions to our responses to other reviewers. We apologize for any inconvenience this may cause. Our responses are as follows:
>
> ***Q1: Definition and Guarantees of Probabilistic Consistency.***
>
> **A1:** In the MSP framework, probabilistic consistency ensures that fine-grained local control remains strictly within the probabilistic boundaries defined by coarse-grained intentions in the continuous latent action space (for formal definitions, please see Reviewer 4 Q1). MSP strengthens Eq. (3) via temporal downsampling ($z^s = \text{Down}(z, r_s)$) and MeanFlow's 1-NFE average velocity field, structurally eliminating quantization errors and structural degradation. Without this, models suffer severe error accumulation in long-horizon tasks. Our ablations confirm this necessity: replacing MeanFlow with diffusion-based or standard flow-matching (Table 3) drops success from 96.8% to 79.2% and 82.2%, respectively. Furthermore, removing intermediate scales (Table 4) plummets performance to 91.6% (and 79.4% for longer sequences). These results unequivocally prove that 1-NFE generation and scale-wise probabilistic modeling are indispensable for cross-scale consistency.
>
> ***Q2: Baseline Comparison and Architectural Differences with H3DP.***
>
> **A2:** Due to character limits, detailed comparisons with H3DP are provided in our response to Reviewer 1 (Q1). We kindly invite you to refer to that section for more details.
>
> ***Q3: Cross-Scale Error Propagation and the Role of Stop-Gradient.***
>
> **A3:** Thank you for raising the critical issue of cross-scale error propagation. Our architecture mitigates this risk by design. During Stage 2 training, the semantic $\hat{z}^s$ depends entirely on the latent variables of previous coarser scales $z^{<s}$ encoded via the Stage 1 VAE framework, rather than predictions from the preceding scale-wise MeanFlow model. This independent optimization removes gradient-level paths for cross-scale error accumulation.
>
> To examine behavior during inference, we measured the Mean Absolute Error (MAE) of latent reconstruction across the coarse-to-fine generation. The MAE remains consistently low and stable across scales: 0.036785, 0.038456, 0.038346, and 0.055440. Although the finest-scale latent shows a slight increase due to its higher dimensionality and sequence length, the overall error remains tightly bounded, indicating no progressive error accumulation.
>
> Regarding the stop-gradient operation $sg(u_{tgt})$ in Eq. (8), its role is not cross-scale decoupling but improving training stability. By blocking gradients through the target average velocity $u_{tgt}$, it avoids double backpropagation through Jacobian-vector products, prevents gradient explosion, and improves training stability at each scale.
>
> ***Q4: Rationale and Ablation of the Scale-Weighting Strategy.***
>
> **A4:** In the MSP architecture, coarser scales correspond to temporally compressed sequences with smaller latent dimensions and are therefore easier to optimize, whereas finer scales must model high-frequency control details over longer horizons with much larger feature spaces. Assigning higher weights to coarse scales would cause the total gradient to be dominated by low-dimensional representations, increasing the risk of underfitting at fine scales.
>
> We performed an ablation study on LIBERO-Long to support this. The proposed scale-increasing weighting achieves a 96.8% success rate, whereas uniform weighting reduces performance to 80.2%, and reverse weighting (assigning higher weights to coarse scales) further lowers it to 75.8%. These results indicate that while coarse scales establish global intentions, finer scales require higher loss weights in long-horizon tasks to ensure precise execution and avoid error accumulation.
>
> ***Q5: VAE Reconstruction Bottleneck and Decoupled Training.***
>
> **A5:** To address this question, we evaluated the reconstruction Mean Absolute Error (MAE) of Stage 1 in the original action space. On the LIBERO benchmark, the MAE is 0.007814, and it remains low at 0.005318 on the more complex bimanual RoboTwin 2.0. Since action commands are normalized to the [-1, 1] range, an absolute error of 0.007814 corresponds to a single-step relative deviation of only 0.391%. These results suggest that Stage 1 does not become a bottleneck for downstream long-horizon control.
>
> Regarding end-to-end joint fine-tuning, preliminary experiments yielded no performance gains, only increased computational costs and instability. We therefore chose decoupled training: Stage 1 rapidly constructs a stable latent manifold from low-dimensional actions, enabling the expensive Stage 2 (vision-action) to train efficiently without disruptive gradient backpropagation, significantly reducing required epochs.

---

> > ### Author Rebuttal · Reviewer_DGyV · 2026-04-02
> >
> > Thank you for your detailed response. My concerns have been addressed and I will keep my positive score.

---

> > > ### Author Response · Authors · 2026-04-03
> > >
> > > Dear Reviewer,
> > >
> > > We deeply appreciate the time and energy you dedicated to reviewing our work! It is wonderful to hear that our rebuttal and new experiments have successfully resolved your concerns.
> > >
> > > Have a fantastic weekend!

---

### Official Review · Reviewer_XNQy · 2026-03-14

**Soundness:** 4
**Presentation:** 4
**Significance:** 3
**Originality:** 3
**Overall Recommendation:** 5
**Confidence:** 4

**Summary:**

This paper proposes Probabilistically Consistent Multi-Scale Action Generation to generate actions from coarse-to-fine in long-horizon imitation learning. The method constructs token stream by downsampling the action latent space. It uses Transformer and a light-weight MeanFlow model to model the multi-scale token stream, enabling consistent action chunk generation and fast inference. Experiments demonstrate SOTA performance and inference speed.

**Compliance With Llm Reviewing Policy:**

Affirmed.

**Final Justification:**

After considering both the paper and the authors’ rebuttal, I maintain my accept recommendation. I find the paper technically strong, well executed, and well supported by extensive empirical results, with a clear practical contribution for long-horizon imitation learning. The proposed coarse-to-fine multi-scale action generation framework is reasonably original, and the paper is generally clear and convincing.

My main questions in the original review were about training dynamics and whether explicit scale information should be provided to MeanFlow. The rebuttal addressed these concerns well by clarifying the joint end-to-end training procedure, explaining the role of causal masking and architectural constraints against information leakage, and providing an additional ablation showing that explicit scale conditioning actually hurts performance. As a result, the rebuttal reinforced my positive assessment rather than changing it substantially.

Overall, I view this as a technically solid and impactful paper with strong experimental support, and I remain **supportive of acceptance**.

**Key Questions For Authors:**

1. How is the Transformer trained? If we train it jointly with MeanFlow, how could we keep it from information leaking, i.e. the MeanFlow module could learn to only rely on the condition. Though the training target has a classfier-free guidance part, the training dynamics should still be clarified and discussed.
2. Also, should the MeanFlow module be aware of the time scale explicitly, i.e. passing $s$ into its condition? I think although MeanFlow could learn from the information in $x^s_t$, we may need to examine the nature of latent space to see differences between $z^s$ and $z^{s+1}$

**Limitations:**

yes

**Strengths And Weaknesses:**

**Strengths:**
- The method is technically sound with frontier generative models and clear formulation.
- The paper is well-supported by extensive experiments and ablations, showing the superior performance in challenging benchmarks as well as real-world scenarios.

**Weaknesses:**
- Although the paper is well-written, I think a pseudo algorithm could be provided in the paper to clarify details of the pipeline.
- Some pipeline designs could be clarified, e.g. how is the Transformer trained (separately or jointly with MeanFlow)? See more in the question part as framework discussions could be as valuable as the method.

---

> ### Author Rebuttal · Authors · 2026-03-30
>
> Dear Reviewer,
>
> We sincerely thank you for your highly inspiring and in-depth technical feedback. To address your concerns, we have further clarified the training dynamics of our model and conducted a dedicated ablation study regarding the time scale conditioning of MeanFlow. Our detailed responses are provided below:
>
> ***Q1: How is the Transformer trained? If we train it jointly with MeanFlow, how could we keep it from information leaking, i.e. the MeanFlow module could learn to only rely on the condition. Though the training target has a classfier-free guidance part, the training dynamics should still be clarified and discussed.***
>
> **A1:** We deeply appreciate you highlighting the need to clarify our training dynamics. In the MSP framework, the scale-wise autoregressive Transformer and the MeanFlow module are jointly trained end-to-end under a unified flow matching objective. To structurally prevent information leakage during this joint optimization, we implement a rigorous scale-wise causal attention mask within the Transformer. This mechanism ensures that when generating semantic conditions for the current scale $s$, the model infers strictly from observed features and coarse-scale history $z^{<s}$, effectively blocking any access to ground-truth action representations of current or finer scales.
>
> Regarding concerns about shortcut solutions, the MeanFlow module is architecturally constrained from bypassing the Transformer. Since the Transformer serves as the exclusive source of structured temporal information, and the MeanFlow loss is computed sequentially at every scale based on prior latent tokens, the network is compelled to learn meaningful hierarchical representations. Furthermore, incorporating Classifier-Free Guidance (CFG) prevents the module from collapsing into a deterministic mapping, as it must simultaneously learn the unconditional velocity field. Finally, to stabilize these overall training dynamics, MSP applies scale-based weights to balance the Transformer's multi-scale representation learning with MeanFlow's probability distribution modeling, effectively avoiding representation collapse.
>
> ***Q2: Also, should the MeanFlow module be aware of the time scale explicitly, i.e. passing into its condition? I think although MeanFlow could learn from the information in $x^s_t$, we may need to examine the nature of latent space to see differences between $z^s$ and $z^{s+1}$ .***
>
> **A2:** We deeply appreciate your insightful observation. In our current implementation, the scale-wise MeanFlow module shares weights across all scales without explicitly receiving the time scale as an independent condition. Instead, it relies on the implicit scale information embedded within the semantic condition $\hat{z}^s$. Since the scale-wise autoregressive Transformer utilizes scale-specific positional encodings and aggregates coarse-scale history $z^{<s}$ when generating $\hat{z}^s$, the feature distributions of $\hat{z}^s$ and $\hat{z}^{s+1}$ are inherently distinct.
>
> To rigorously verify your hypothesis, we conducted an additional ablation study by explicitly introducing time scale information into the MeanFlow condition. Specifically, the sequence length $t_s$ was mapped to a scale embedding and concatenated with $\hat{z}^s$ and timestep $t$ as input to our lightweight three-layer MLP ResNet. However, we observed that adding explicit time scale information caused the average success rate on LIBERO-Long to drop from 96.8% to 87.8%.
>
> Our analysis suggests several primary reasons for this degradation. First, it stems from feature redundancy; since the scale-wise autoregressive Transformer already processes $\hat{z}^s$ using rotary positional encodings and scale-wise causal attention masks, $\hat{z}^s$ already contains rich, context-integrated scale information. Forcing an explicit time scale embedding introduces redundancy that interferes with high-order feature extraction. Second, to maintain inference efficiency, MeanFlow employs a lightweight three-layer MLP ResNet. Explicitly introducing extra dimensions disrupts the representational balance, increasing optimization difficulty and leading to higher cumulative errors in long-horizon tasks. Finally, implicit conditioning facilitates smoother transitions between scales, whereas hard, explicit features may introduce unnecessary boundaries within the latent space, potentially undermining cross-scale probabilistic consistency.
>
> We hope the response has addressed your concerns. If you have any additional concerns or comments that we may have missed in our responses, we would be most grateful for any further feedback from you to help us further enhance our work. We are more than happy to include all these discussions in the camera-ready version of this work.

---

> > ### Author Rebuttal · Reviewer_XNQy · 2026-04-01
> >
> > Thank you for your detailed replies. My concerns have been addressed meaningfully. I believe the paper is technically sound and I will keep my score.

---

> > > ### Author Response · Authors · 2026-04-03
> > >
> > > Dear Reviewer,
> > >
> > > Thank you for your time and effort in reviewing our paper! We are very glad that our explanations and additional results have effectively addressed your questions and concerns.
> > >
> > > Wishing you a wonderful weekend!

---

### Official Review · Reviewer_6Wd2 · 2026-03-17

**Soundness:** 3
**Presentation:** 3
**Significance:** 4
**Originality:** 3
**Overall Recommendation:** 5
**Confidence:** 5

**Summary:**

This paper proposes MSP, a coarse-to-fine policy for long-horizon imitation learning in continuous action spaces. The method first encodes action sequences into a continuous latent space with a variational autoencoder (VAE), then builds a temporal multi-scale hierarchy by downsampling that latent sequence. A scale-wise autoregressive Transformer predicts semantic conditions at each scale, and a MeanFlow module generates the corresponding latent variables so that finer-scale actions remain probabilistically consistent with coarser plans. Across simulation and real-world manipulation tasks, the paper shows improved performance.

**Compliance With Llm Reviewing Policy:**

Affirmed.

**Final Justification:**

My overall recommendation is accept. So I maintain this. I increased my score on originality as I am now more convinced about this aspect of the work.

**Key Questions For Authors:**

1) The paper discusses H3DP in related work, but it is not included in the baseline set. Could the authors compare against H3DP directly, or explain clearly why such a comparison was not feasible? Since MSP’s main claim is about coarse-to-fine consistency across scales, a comparison to a directly relevant hierarchical diffusion baseline would materially strengthen the paper’s empirical case. My evaluation would improve if the authors can provide either such a comparison or a convincing explanation.

2) RoboTwin2.0 contains 50 tasks, but the paper evaluates only 8 representative tasks. How were these tasks selected, and do the conclusions hold on a larger subset or on the full benchmark? The current results are promising, but broader coverage would make the generalization claim more convincing and reduce concern about selection effects.

3) In the real-world section, the comparisons are only against DP and DSP. Could the authors clarify why stronger or more directly matched baselines were not included, and whether additional real-robot comparisons were attempted? The real-world results are a strength of the paper, but the baseline set there is relatively narrow.

4) Have the authors considered evaluating MSP on an additional standard long-horizon manipulation benchmark beyond LIBERO and RoboTwin2.0? I do not view this as strictly necessary for acceptance, but it would make the case for generality substantially stronger, especially since the paper aims to address long-horizon action generation rather than only one benchmark family.

**Limitations:**

I suggest that the authors explicitly discuss: (1) the limited benchmark coverage relative to the paper’s broad claims, including only 8 RoboTwin2.0 tasks and a narrow real-world baseline set; (2) possible failure modes in long-horizon real-robot deployment, such as compounding errors, recovery failure, and robustness under distribution shift, and 3) finally future extensions and aspects that need further investigation.

**Strengths And Weaknesses:**

Soundness: The paper is technically solid overall. the method is well motivated, the architecture is intuitive, and the empirical study covers simulation, few-shot transfer, and real-world long-horizon tasks, with strong results on LIBERO and RoboTwin2.0 plus useful ablations on the main design choices. My main weakness is that the experimental case could be stronger against additional directly relevant baselines, especially hierarchical diffusion-style methods discussed in related work, since the current comparison set does not fully test the paper’s core claim about cross-scale consistency.

Presentation: The paper is clearly written and easy to follow at a high level. The two-stage structure, Figure 2, and the coarse-to-fine narrative make the main idea accessible, and the method is positioned reasonably well relative to discrete, autoregressive, and diffusion/flow baselines. The main presentation weakness is that some distinctions between closely related hierarchical methods are still a bit insufficient, so the paper would benefit from a stronger explanation of exactly how MSP differs from the most relevant prior multiscale alternatives and why some of those methods were not included experimentally.

Significance: I view the paper as significant because long-horizon imitation learning in continuous action spaces is an important problem, and the reported gains are strongest precisely on the harder long-horizon settings where error accumulation matters most. The real-world results and few-shot transfer experiments also make the work feel practically relevant rather than purely benchmark-driven.

Originality: The paper’s novelty comes less from inventing an entirely new primitive and more from combining several ideas in a thoughtful way such as continuous latent action modeling, explicit multiscale hierarchy, scale-wise autoregressive semantic conditioning, and MeanFlow-based one-step generation. I think this combination is interesting and produces a coherent perspective on cross-scale consistency. The weakness is that the paper’s closest conceptual neighbors are also quite recent, so the originality claim would be more convincing with stronger empirical separation from the most relevant hierarchical prior work, not just a conceptual distinction in the related-work section.

---

> ### Author Rebuttal · Authors · 2026-03-30
>
> Dear Reviewer,
>
> We sincerely value the time and profound insights you dedicated to evaluating our work. Due to strict character limits, we have summarized your questions as Q1–Q4 to prioritize a comprehensive response. Our answers correspond to your points in order:
>
>
> ***Q1: Comparison with H3DP.***
>
> **A1:** We deeply appreciate you bringing this excellent related work to our attention. Since H3DP's code is closed-source, to avoid unofficial reimplementation bias, we evaluated MSP directly on their RoboTwin1.0 benchmark. Under perfectly aligned settings (identical camera views and expert demonstrations), MSP significantly outperforms H3DP by +12.25% in average success rate across 8 tasks, as shown in R1-Table1.
>
> > R1-Table1: Performance comparison between H3DP and MSP on RoboTwin1.0.
>
> | Task                     | H3DP (%) |   MSP (%) |
> | :----------------------- | -------: | --------: |
> | Apple Cabinet Storage    |       98 |   **100** |
> | Dual Bottles Pick (Easy) |       48 |    **93** |
> | Dual Bottles Pick (Hard) |   **53** |        51 |
> | Block Handover           |       70 |    **95** |
> | Block Hammer Beat        |   **85** |        81 |
> | Diverse Bottles Pick     |       25 |    **51** |
> | Pick Apple Messy         |   **35** |        27 |
> | Tool Adjust              |       45 |    **59** |
> | **Average**              |    57.38 | **69.63** |
>
>
>
> ***Q2: Evaluation on additional RoboTwin2.0 tasks.***
>
> **A2:** Thank you for highlighting our model's generalizability and comprehensive evaluation. Originally, we selected 8 tasks from RoboTwin2.0 across four temporal horizon levels (short, medium, long, and extra-long) based on execution time. To further demonstrate MSP's broad applicability and superiority, we expanded our evaluation with 8 new RoboTwin2.0 tasks covering varying horizons (ranging from 85 to 537 steps). We compared MSP against the Diffusion Policy (DP) baseline, with results and sequence lengths detailed in **R1-Table2**:
>
> > **R1-Table2:** Performance comparison between DP and MSP on additional RoboTwin2.0 tasks.
>
> | Task (Steps)                | DP (%) |  MSP (%) |
> | :-------------------------- | -----: | -------: |
> | click alarmclock (85)       |     77 |   **98** |
> | beat block hammer (113)     | **72** |       70 |
> | place bread basket (231)    |     26 |   **38** |
> | open laptop (258)           | **82** |       80 |
> | dump bin bigbin (265)       |     85 |   **93** |
> | place cans plasticbox (289) |     48 |   **91** |
> | hanging mug (340)           |     17 |   **27** |
> | open microwave (537)        |     61 |   **99** |
> | **Average**                 |   58.5 | **74.5** |
>
>
>
> ***Q3: Baseline selection in real-world experiments.***
>
> **A3:** We are grateful for your insightful questions concerning the real-world baselines. For our real-world evaluations, we selected DP as the foundational baseline due to its widespread adoption in the field. Additionally, we included the recently proposed DSP (2025) because it shares conceptual similarities with our multi-scale paradigm and achieved the second-best performance (behind MSP) across our simulation benchmarks, making it a highly competitive baseline for hardware comparison. Regarding H3DP, the absence of an official open-source codebase prevented us from objectively reproducing and rigorously testing it in real-world environments. Finally, while we initially conducted preliminary experiments on a UR5e, frequent triggering of its low-level overheating protection restricted us from performing large-scale, long-horizon systematic evaluations. We plan to thoroughly verify our model's cross-embodiment generalizability on a broader range of hardware in future work.
>
>
>
> ***Q4: Suggestion to expand evaluation benchmarks.***
>
> **A4:** We are truly grateful for the suggestion to expand our evaluation benchmarks. For RoboTwin1.0, please refer to our detailed results in Response A1. Additionally, we had previously evaluated our model on the Push-T simulation benchmark. We initially omitted these results from the main manuscript due to strict page limits and because Push-T is not a long-horizon task (which is our primary focus). Nevertheless, Push-T rigorously tests a model's ability to capture complex contact dynamics and multimodal action distributions. Under identical conditions, MSP achieves a success rate of 94.2% on Push-T, outperforming the strong DP-C baseline (91.0%). While configuring an entirely new long-horizon environment from scratch within the short rebuttal window is temporally prohibitive, we believe our comprehensive evaluations across LIBERO, RoboTwin2.0, RoboTwin1.0, and Push-T robustly validate the generalization capabilities of our approach.
>
> We hope we have fully resolved your concerns. If any questions remain, we welcome your further feedback to help us improve our work. We will gladly incorporate all these discussions into the camera-ready revision.

---

> > ### Author Rebuttal · Reviewer_6Wd2 · 2026-04-03
> >
> > Most of my concerns are addressed.

---

> > > ### Author Response · Authors · 2026-04-04
> > >
> > > Dear Reviewer,
> > >
> > > We deeply appreciate the time and energy you dedicated to reviewing our work! It is wonderful to hear that our detailed explanations have successfully resolved your concerns. We also sincerely thank you for your insightful suggestions, which have significantly improved our paper.
> > >
> > > We hope you are having a good week.

---

### Decision · Program_Chairs · 2026-04-30

**Decision:**

Accept (spotlight)

**Comment:**

This paper proposes MSP, a coarse-to-fine framework for long-horizon robotic imitation learning that enforces probabilistic consistency across temporal scales via VAE-based latent compression, scale-wise autoregressive conditioning, and MeanFlow-based one-step generation. All four reviewers are positive (scores: 5, 5, 4, 4), and all marked their concerns as fully resolved after the rebuttal. The reviewers consistently praised the clean two-stage design, strong empirical results across multiple benchmarks (LIBERO, RoboTwin 1.0/2.0, real-world), informative ablation studies, and practical efficiency. Initial concerns regarding the missing H3DP comparison, formal definition of probabilistic consistency, cross-scale error accumulation, and scale loss weighting were thoroughly addressed with new experiments and analysis. The paper makes a solid contribution to long-horizon action generation with a well-executed combination of recent generative modeling techniques.